Manuscript prepared for The Cryosphere
with version 2014/05/22 6.83 Copernicus papers of the LaTeX class copernicus.cls.
Date: 5 December 2016

# Snow fracture in relation to slab avalanche release: critical state for the onset of crack propagation

**Johan Gaume[1,2], Alec van Herwijnen[1], Guillaume Chambon[3], Nander Wever[1,2], and Jürg Schweizer[1]**

[1]WSL Institute for Snow and Avalanche Research SLF, Davos, Switzerland
[2]EPFL Swiss Federal Institute of Technology, School of Architecture, Civil and Environmental Engineering, Lausanne, Switzerland.
[3]Université Grenoble Alpes, Irstea, UR ETGR, Grenoble, France

*Correspondence to:* Johan Gaume (johan.gaume@epfl.ch)

**Abstract.** The failure of a weak snow layer buried below cohesive slab layers is a necessary, but insufficient condition for the release of a dry-snow slab avalanche. The size of the crack in the weak layer must also exceed a critical length to propagate across a slope. In contrast to pioneering shear-based approaches, recent developments account for weak layer collapse and allow for better explaining typical observations of remote triggering from low-angle terrain. However, these new models predict a critical length for crack propagation that is almost independent of slope angle, a rather surprising and counterintuitive result. Based on discrete element simulations we propose a new analytical expression for the critical crack length. This new model reconciles past approaches by considering for the first time the complex interplay between slab elasticity and the mechanical behaviour of the weak layer including its structural collapse. The crack begins to propagate when the stress induced by slab loading and deformation at the crack tip exceeds the limit given by the failure envelope of the weak layer. The model can reproduce crack propagation on low-angle terrain and the decrease in critical length with increasing slope angle as modeled in numerical experiments. The good agreement of our new model with extensive field data and the ease of implementation in the snow cover model SNOWPACK opens promising prospect towards improving avalanche forecasting.

## 1 Introduction

Snow slab avalanches range among the most prominent natural hazards in snow covered mountainous regions throughout the world. The winter 2014/2015 served as a cruel reminder of the destructive power of this ubiquitous natural hazard with 132 fatalities, just for the European Alps. The ability to reliably forecast avalanche danger is therefore of vital importance and requires a sound understanding of avalanche release processes.

Avalanches are the result of numerous factors and processes interacting over a large range of temporal and spatial scales (Schweizer et al., 2003). While snow slab avalanches can come in many different sizes, from a few meters to several kilometers, they initiate within the snow cover by local damage processes at the grain scale. Indeed, the release of a dry-snow slab avalanche (Fig. 1a) requires the formation of a localized failure within a so-called weak layer (WL) buried below cohesive slab layers (Fig. 1b). The initial failure resulting in a macroscopic crack in the WL develops from micro-scale heterogenities by damage accumulation (Schweizer et al., 2008; Gaume et al., 2014b), or directly below a local overload such as a skier or a snowmobile (van Herwijnen and Jamieson, 2005; Thumlert and Jamieson, 2014). Stress concentrations at the crack tips will then determine if crack propagation and eventually slope failure occurs (McClung, 1979; Schweizer et al., 2003), even if the average overlying stress is lower than the average weak layer strength (knock-down effect, Fyffe and Zaiser, 2004; Gaume et al., 2012, 2013, 2014b). The size of the initial crack at which rapid crack propagation occurs is called the critical crack length and represents an instability criterion for material failure (Anderson, 2005). It is a crucial variable to evaluate snow slope instability (Reuter et al., 2015).

Information on snow cover stratigraphy, especially the presence and characteristics of WLs and the overlying slab, is thus essential for avalanche forecasting. Traditionally, such information is obtained through manual snow cover observa-

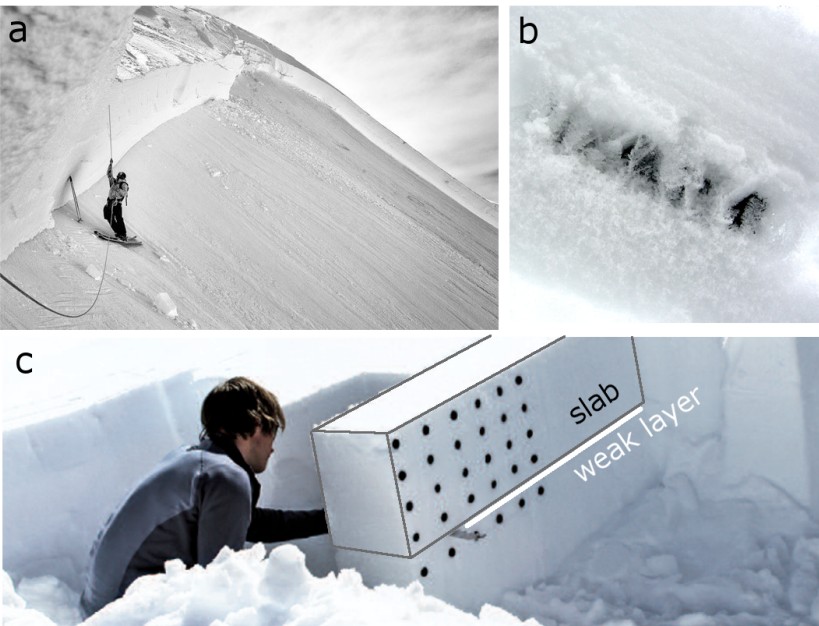

Figure 1: (a) Crown fracture of a dry-snow slab avalanche in Mt Baker, USA ©Grant Gunderson. (b) Surface hoar weak layer. (c) Propagation Saw Test. The weak layer is represented in white, the slab in grey. The black dots are markers used for particle tracking to measure slab deformation.

tions, such as snow profiles and stability tests (Schweizer and Jamieson, 2010). However, these observations are time consuming, somewhat subjective and only provide point observations. Snow cover models such as CROCUS (Brun et al., 1992) or SNOWPACK (Lehning et al., 1999) provide a valuable alternative to obtain more highly resolved snow stratigraphy data. However, to evaluate snow slope instability based on model output, avalanche formation processes are greatly simplified, and reduced to accounting for the balance between shear strength of the WL and shear stress due to the weight of the overlying slab, sometimes including a skier overload (Schweizer et al., 2006; Monti et al., 2016). This 'strength-over-stress' approach is only relevant for failure initiation and does not account for crack propagation, the second fundamental process in avalanche release.

Due to the very complex nature of crack propagation in multilayered elastic systems under mixed-mode loading, theoretical and analytical approaches are not yet conceivable (Hutchinson and Suo, 1992). In the past, simplifying assumptions have been used to propose analytical models for the critical crack length. For instance, McClung (1979); Chiaia et al. (2008) and Gaume et al. (2014b) assumed a weak layer without thickness which allowed solution to the problem in the down-slope direction only, by neglecting the effect of the volumetric collapse of the weak layer as e.g. described by Jamieson and Schweizer (2000). On the other hand, Heierli et al. (2008) assumed a weak layer of finite thickness with a slope-independent failure criterion and a completely rigid behavior allowing to neglect the elastic mismatch between the slab and the weak layer. With the development of new field tests, in particular the propagation saw test (PST, Fig. 1c) (van Herwijnen and Jamieson, 2005; Gauthier and Jamieson, 2006; Sigrist and Schweizer, 2007), it is now possible to directly evaluate the critical crack length, and thus determine crack propagation propensity. Particle tracking velocimetry (PTV) analysis of PSTs has highlighted the importance of the elastic bending of the slab induced by the loss of slab support due to weak layer failure (induced by a saw) prior to crack propagation (van Herwijnen et al., 2010, 2016; van Herwijnen and Birkeland, 2014). To include slab bending in the description of slab avalanche release mechanisms, Heierli et al. (2008) proposed the anticrack model. This model provides an analytical framework to estimate the critical crack length as a function of slab properties (thickness, density and elastic modulus) and the WL specific fracture energy, a WL property quantifying the resistance to crack propagation. While some crucial features of the mechanical behavior of the WL, including elasticity and shape of the failure envelope are not included, the anticrack model provides a significant step forward as it accounts for various aspects that were left unexplained by previous theories, such as crack propagation on low-angle terrain and remote triggering of avalanches.

To evaluate the critical crack length based on the anticrack model, the WL specific fracture energy is required. Presently, it can be estimated using three existing methods: (i) through PTV or finite element analysis of the PST (Sigrist and Schweizer, 2007; van Herwijnen et al., 2010, 2016; Schweizer et al., 2011); (ii) from snow micro-penetrometer

(SMP) measurements (Schneebeli et al., 1999) by integrating the penetration resistance over the thickness of the WL (Reuter et al., 2015) and (iii) from X-ray computer tomography-based (CT) microstructural models (LeBaron and Miller, 2014). Depending on the method, estimates of the WL specific fracture energy can differ by as much as two orders of magnitude, resulting in widely different values of the critical crack length. Strength-of-material approaches have also been developed to evaluate the conditions for the onset of crack propagation (Chiaia et al., 2008; Gaume et al., 2013, 2014b). These methods require WL strength, a property which is more readily measurable (Jamieson and Johnston, 2001), rather than the specific fracture energy. Yet, in contrast to the anticrack model, the latter strength-of-material approaches do not account for slab bending which leads to additional stress concentrations, hence these models tend to overestimate the critical length.

Clearly, the various methods to estimate the critical crack length all have their respective shortcomings, and a unified approach which incorporates all relevant processes is thus far missing. To overcome these limitations and take into account all the important physical ingredients, we propose to evaluate the critical crack length for different snowpack stratigraphies using discrete element (DEM) simulations. Similar to the field experiments, in the simulations we gradually create a crack in the WL with a saw until rapid propagation occurs (Fig. 2). On the basis of our numerical results, we then introduce a new expression for the critical crack length which accounts, for the first time, for the complex interplay between loading, elasticity, failure envelope of the WL and its structural collapse. The predictive capabilities of this new expression, with respect to field data, are discussed and compared to previous models.

## 2   Methods

### Discrete element model

We model crack propagation in a slab-WL system using the discrete element method (DEM). DEM is well suited to represent large deformations as well as the evolution of the microstructure of materials in a dynamic context (Radjai et al., 2011; Hagenmuller et al., 2015; Gaume et al., 2011, 2015b). The simulations are performed using PFC2D (by Itasca), implementing the original soft-contact algorithm of Cundall and Strack (1979). The numerical setup and the cohesive contact law implemented is fully described in Gaume et al. (2015b). We recall here the main characteristics of the DEM model.

The simulated system (Fig. 2a) is 2D and composed of a fixed substratum, a WL of thickness $D_{wl}$ (varied between 0.02 and 0.06 m) and a slab of thickness $D$ (varied between 0.2 and 0.8 m). The slab is modeled with spherical elements of radius $r = 0.01$ m with a square packing. As explained

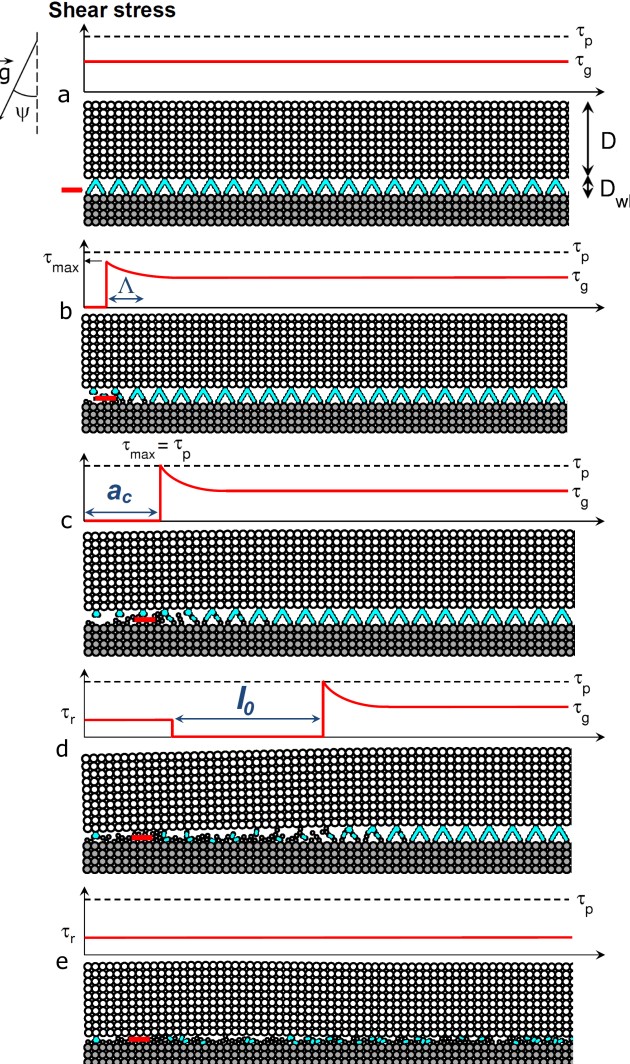

Figure 2: Successive snapshots (a to e) of a DEM simulation of the propagation saw test (PST). The plots on top of each snapshot represent illustrations of the shear stress $\tau$ (red line) in the WL. $D$ is the slab thickness (slope normal), $D_{wl}$ is the WL thickness, $\psi$ is the slope angle, $\tau_{max}$ is the maximum shear stress at the crack tip, $\tau_p$ is the WL shear strength (dashed line), $\tau_g = \rho g D \sin\psi$ is the shear stress due to the slab weight and $\tau_r$ is the residual frictional stress. $a_c$ is the critical crack length, $\Lambda$ is the characteristic lengthscale of the system and $l_0$ is the touchdown length (see Sec. 3). The red segment represents the saw used to cut inside the weak layer.

in Gaume et al. (2015b), these elements are not intended to represent the real snow grains. They constitute entities of discretization used to model an elastic continuum of density $\rho$, Young's modulus $E$ and Poisson's ratio $\nu$. The WL is composed of elements of radius $r_{wl} = r/2$ with a packing of collapsible triangular shapes of the same size as the WL thickness (Fig. 2a) aimed at roughly representing the porous mi-

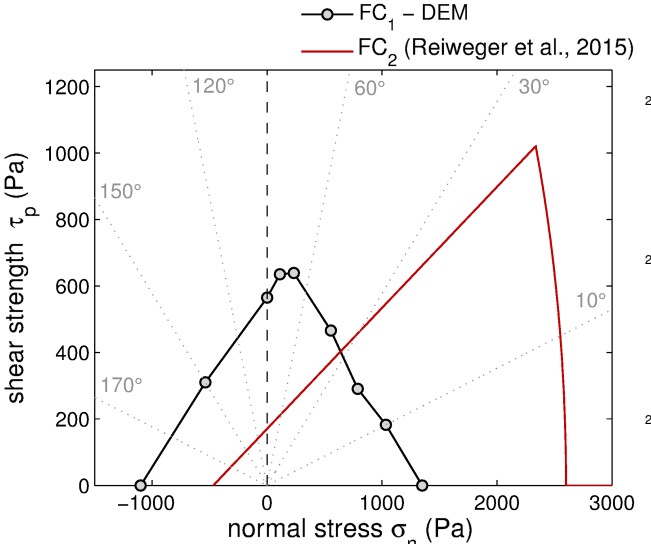

Figure 3: Failure criterion $FC_1$ of our modeled weak layer (black circles) obtained from mixed-mode shear-compression loading tests. $FC_2$ is the high-rate mixed-mode failure envelope found by Reiweger et al. (2015). The grey dotted lines represent angles of loading $\psi$ such as $\tan\psi = \tau_g/\sigma_n$ where $\tau_g$ is the shear stress. Compression corresponds to positive values of $\sigma_n$.

crostructure of persistent WLs such as surface hoar (Fig. 1b) or depth hoar.

We used the cohesive contact law detailed in Gaume et al. (2015b). The bonds are characterized by specific elasticity and strength parameters which have been calibrated to obtain the desired macroscopic (bulk) properties. For the slab, numerical biaxial tests were performed to characterize the macroscopic Young's modulus $E$ as a function of micro-mechanical parameters. For the WL, mixed-mode shear-compression loading simulations were performed to determine the failure envelope (Fig. 3). Through the triangular shape of the WL structure, the main features of real WL failure envelopes (Chandel et al., 2014; Reiweger et al., 2015) are captured, notably the closed envelope necessary to obtain failures both in shear and compression.

The applied loading represents a typical experimental setup of a PST (van Herwijnen and Jamieson, 2005; Gauthier and Jamieson, 2006; Sigrist and Schweizer, 2007). It consists of a combination of gravity (slope angle $\psi$) and advancing a rigid "saw" (in red in Fig. 2) at a constant velocity $v_{saw} = 2$ m/s through the WL. The saw thickness is $h_{saw} = 2$ mm and the length of the system is $L = 2$ m (Bair et al., 2014; Gaume et al., 2015b).

## Comparison with propagation saw test (PST) experiments

The dataset consists of 93 PST experiments which were presented in Gaume et al. (2015b). It includes the average slab density $\rho$, slab thickness $D$, slope angle $\psi$ and WL thickness $D_{wl}$. The WL specific fracture energy $w_f$ was evaluated from the penetration resistance of the weak layer using the snow micro-penetrometer (SMP) according to Reuter et al. (2015) and ranges from 0.07 to 2.9 J/m$^2$. Reuter et al. (2015) showed a good correlation between the SMP-derived $w_f$ and that derived using particle tracking and the anti-crack model (van Herwijnen et al., 2016). The shear strength $\tau_p$ of the WL was not measured but we used the mixed-mode shear-compression failure envelope defined by Reiweger et al. (2015) based on laboratory experiments. This failure envelope (in red in Fig. 3), i.e. the relation between the shear strength $\tau_p$ and the slope normal stress $\sigma_n$, is described by the following Mohr-Coulomb-Cap model:

$$\tau_p = \tau_p^{mc} = c + \sigma_n \tan\phi \quad \text{for} \quad \psi > \psi_t, \tag{1}$$

$$\tau_p = \tau_p^{cap} = b\sqrt{1 - \frac{(\sigma_n + \sigma_t)^2}{(\sigma_c + \sigma_t)^2}} \quad \text{for} \quad \psi < \psi_t. \tag{2}$$

where $\psi_t = 23°$ is the angle corresponding to a transition between the Mohr-Coulomb and the cap regimes, $c$ is the cohesion, $\phi = 20°$ is the friction angle, $\sigma_t = c\tan\phi$ is the tensile strength, $\sigma_c = 2.6$ kPa is the compressive strength and

$$b = K\sqrt{\frac{(\sigma_t + \sigma_c)^2}{(\sigma_t + \sigma_c)^2 - (\frac{K}{\tan\phi})^2}}. \tag{3}$$

$K = 1$ kPa is the maximum shear strength (Reiweger et al., 2015). The cohesion $c$ (shear strength for $\sigma_n = 0$) can be derived from the WL specific fracture energy $w_f$ using the results of Gaume et al. (2014b):

$$c = \frac{\sqrt{2DE'w_f}}{2\Lambda}, \tag{4}$$

where $\Lambda$ is a characteristic lengthscale of the system (see Sec. 3 and Gaume et al., 2013, 2014b). Note that, for the 93 PST experiments, the slope normal stress $\sigma_n$ was lower than 2 kPa and thus only the Mohr-Coulomb part of the failure envelope (Eq. 1) was used to compute the shear strength $\tau_p$. For these stress states (low slope normal stress), Reiweger et al. (2015) showed that failure was almost independent of the loading rate (in the brittle range) and thus independent of fast sintering effects (Szabo and Schneebeli, 2007).

The Young's modulus of the slab $E$, which was not measured, was derived from density according to Scapozza (2004):

$$E = 5.07 \times 10^9 \left(\frac{\rho}{\rho_{ice}}\right)^{5.13}, \tag{5}$$

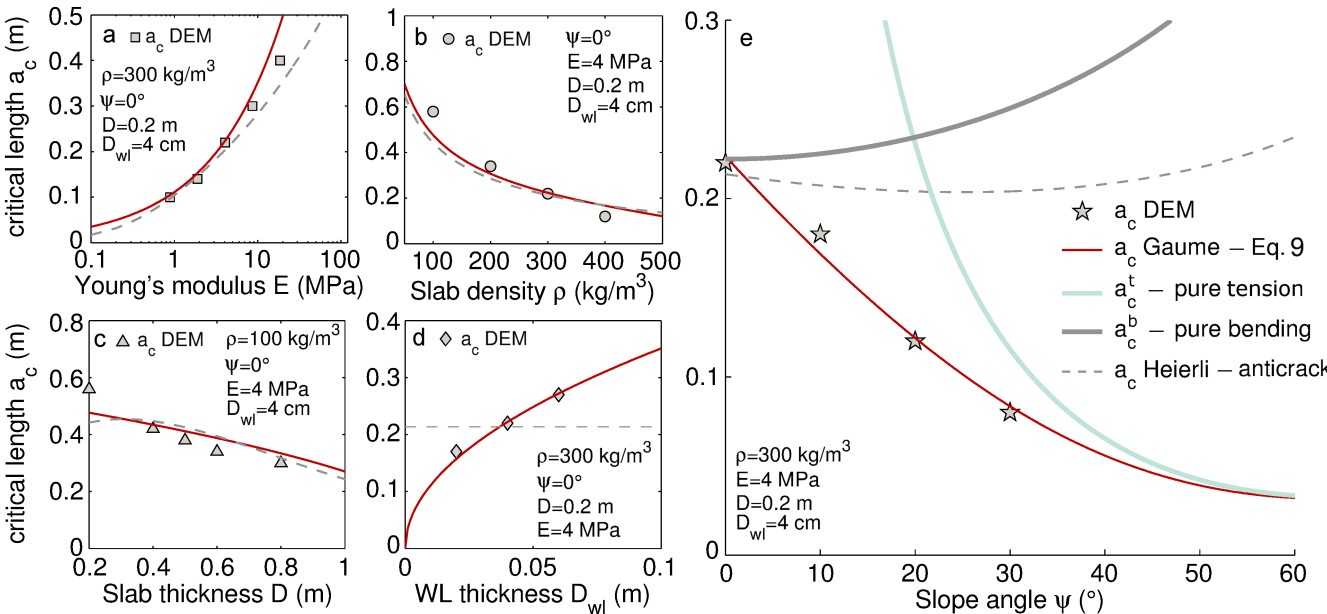

Figure 4: Critical length $a_c$ for crack propagation as a function of (a) Young's modulus $E$ of the slab, (b) slab density $\rho$, (c) slab thickness $D$, (d) WL thickness $D_{wl}$ and (e) slope angle $\psi$. The symbols represent the critical length obtained from the DEM simulations and the solid lines represent the critical length modeled from Eq. 9 and for different failure behaviors. Dashed lines indicate the critical length obtained with the anticrack model (Heierli et al., 2008) assuming $w_f = 0.1$ J/m$^2$.

with $\rho_{ice} = 917$ kg/m$^3$. The WL shear modulus $G_{wl}$ was taken constant equal to 0.2 MPa according to the laboratory experiments performed on snow failure by Reiweger et al. (2010) and Poisson's ratio of the slab $\nu$ was taken equal to 0.2 (Mellor, 1975; Podolskiy et al., 2013).

## 3 Results

### DEM simulations

In the simulations, the crack of length $a$ created by the advancing saw in the WL induces slope-parallel and slope-normal displacements of the slab due to gravity leading to tension and bending in the slab. This results in stress concentrations at the crack tip where the shear stress $\tau = \tau_{max}$ is maximum and larger than the shear stress due to slab weight $\tau_g$. The critical crack length $a_c$ required for the onset of dynamic crack propagation in the WL is reached when $\tau_{max}$ meets the shear strength $\tau_p$ (Fig. 2c).

We performed a series of systematic simulations to investigate the influence of snow cover parameters on $a_c$ (Fig. 4). Slab properties (slab density $\rho$, slab elastic modulus $E$, slab thickness $D$), WL thickness $D_{wl}$ and slope angle $\psi$ were varied independently in the simulations. Overall, $a_c$ was found to increase with increasing elastic modulus of the slab $E$ and with WL thickness $D_{wl}$. On the contrary, $a_c$ decreased with increasing slab density $\rho$, with increasing slab thickness $D$ and with increasing slope angle $\psi$.

### Analytical expression for the critical crack length

The discrete element simulations revealed that the maximum shear stress at the crack tip can be decomposed into two terms related, to slab tension ($\tau_{max}^t$) and slab bending ($\tau_{max}^b$):

$$\tau_{max} = \tau_{max}^t + \tau_{max}^b. \tag{6}$$

When disregarding slab bending (weak layer with no thickness), the maximum stress $\tau_{max}^t$ depends on the shear stress due to the weight of the slab $\tau_g$, the crack length $a$ and a characteristic lengthscale of the system $\Lambda$ (Chiaia et al., 2008; Gaume et al., 2013, 2014b):

$$\tau_{max}^t = \tau_g \left(1 + \frac{a}{\Lambda}\right) \tag{7}$$

The lengthscale $\Lambda$ represents the characteristic scale of the exponential decay of the shear stress $\tau$ close to the crack tip (Fig. 2b). It is given by $\Lambda = (E'DD_{wl}/G_{wl})^{1/2}$ where $E' = E/(1 - \nu^2)$ is the plane stress elastic modulus of the slab and $G_{wl}$ the WL shear modulus (Gaume et al., 2013). We assume the shear stress inside the WL to be equal to the gravitational stress acting at the slab - WL interface, i.e. $\tau_g = \rho g D \sin\psi$. Note that in the limiting case of a WL with zero thickness ($D_{wl} \to 0$), the characteristic length is defined as $\Lambda = (E'D/k_{wl})^{1/2}$, with $k_{wl}$ the shear stiffness of the interfacial WL. Hence, as in the anticrack model (Heierli et al., 2008) (where WL failure is considered as an interfacial fail-

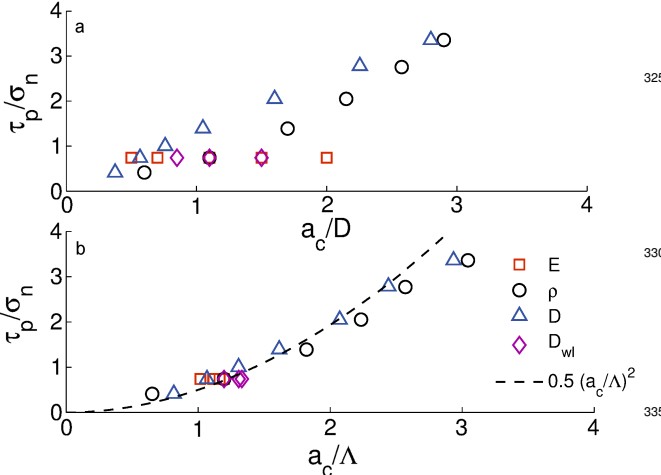

Figure 5: Ratio between the shear strength $\tau_p$ and the slope normal stress $\sigma_n$ versus the ratio between the critical length $a_c$ and (a) slab thickness $D$ or (b) characteristic length $\Lambda$ (b) for flat terrain ($\psi = 0°$, i.e. $\tau_g = 0$). The symbol/color in the legend indicates the parameter which was varied in the DEM simulations. The dashed line corresponds to Eq. 8.

ure), WL thickness $D_{wl}$ plays no role in the model for a constant WL stiffness $k_{wl}$.

The tension term alone is unable to predict stress concentrations and thus crack propagation on flat terrain ($\psi = 0$), a process that exists, exemplified by numerous field observations (Johnson et al., 2004; van Herwijnen and Jamieson, 2007) and our DEM simulations (Fig. 4e). To resolve this discrepancy, the second term in Eq. 6 accounts for slab bending induced by WL collapse. Our DEM simulations showed that this term depends on the slope normal stress $\sigma_n$ and the ratio $a/\Lambda$ (Fig. 5b) and can be expressed as:

$$\tau_{max}^b \approx \frac{1}{2}\sigma_n\left(\frac{a}{\Lambda}\right)^2 \tag{8}$$

For the comparison with the anticrack model which assumes a rigid weak layer, one can consider the bending of a beam over a rigid foundation (Timoshenko and Goodier, 1970). In this case, the bending term $\tau_{max}^b$ would scale with $\sigma_n(a/D)^2$, independent of the elastic properties of the slab and the WL (similar to the anticrack model). In the present formulation, scaling with $a/\Lambda$ instead of $a/D$ provides a means to account for the elastic mismatch between the slab and the WL and to adequately reproduce the numerical results (Fig. 5).

From Eq. 6 the critical length can be obtained by solving $\tau_{max} = \tau_p$ where $\tau_p$ is the shear strength given by the failure envelope of the material (Gaume et al., 2015b; Reiweger et al., 2015):

$$a_c = \Lambda\left[\frac{-\tau_g + \sqrt{\tau_g^2 + 2\sigma_n(\tau_p - \tau_g)}}{\sigma_n}\right] \tag{9}$$

Theoretically, this expression is valid only if crack propagation occurs before the slab touches the broken WL, i.e. if the vertical displacement induced by bending remains lower than the collapse height $h_c$. The length $l_0$ (Fig. 2d) required for the slab to come into contact with the broken WL can be expressed using beam theory: $l_0 = \left(\frac{2ED^2 h_c}{3\rho g \cos\psi}\right)^{1/4}$ (Gaume et al., 2015b). For realistic model parameters, $a_c$ was always substantially lower than $l_0$ (not shown).

The agreement between Eq. 9 and results from the DEM simulations is excellent (red solid lines in Fig. 4). We emphasize that scaling of $\tau_{max}^b$ with $a/\Lambda$ is of critical importance. It also provides an explanation for the gentler decrease of $a_c$ with $D$ compared to $\rho$, even though $D$ and $\rho$ equally contribute to the load. Indeed, for a constant load, thicker slabs will result in lower stress concentrations at the crack tip (Eq. 6) due to an increase of $\Lambda$.

The predictions of Eq. 9 also compare well with results obtained from 93 PST experiments (Fig. 6). Overall, our model provides very good estimates of the measured critical crack lengths, as demonstrated by the proximity of the data to the 1:1 line despite substantial scatter ($R^2 = 0.58$). As for the simulations, the critical length in PSTs was always lower than the length $l_0$ (not shown).

## 4   Discussion

### Comparison with the anticrack model

We compare how well our new analytical expression (Eq. 9) and the anticrack model (Heierli et al., 2008) can reproduce the dependence of the critical crack length on system properties as obtained with our DEM simulations (Fig. 4). The anticrack model reproduces the influence of $E$, $\rho$ and $D$ on $a_c$ well for $\psi = 0$, although less accurately than Eq. 9. However, the influence of WL thickness $D_{wl}$ and slope angle $\psi$ on $a_c$ was very poorly reproduced by the anticrack model, both in terms of absolute values and trends. In particular, a slope angle $\psi > 0$ would lead to similar trends of $a_c$ with $E$, $\rho$ and $D$ but with overestimated values.

The decrease of $a_c$ with slope angle, observed in our DEM results and predicted by Eq. 9, is of particular interest. This trend is in clear contradiction with one of the main outcomes of the anticrack model (Heierli et al., 2008), namely that the critical length is almost independent of slope angle. The discrepancy arises from the fact that the anticrack model (i) assumes that the failure behaviour of the WL is slope independent, (ii) disregards WL elasticity, and (iii) does not adequately account for the interplay between tension and bending in the slab as also shown in van Herwijnen et al. (2016). Concerning WL thickness, a thin WL leads to higher stress concentrations in bonds between the grains and thus to a smaller critical crack length (Fig. 4d). This effect cannot be reproduced by the anticrack model due to the rigid character of the WL.

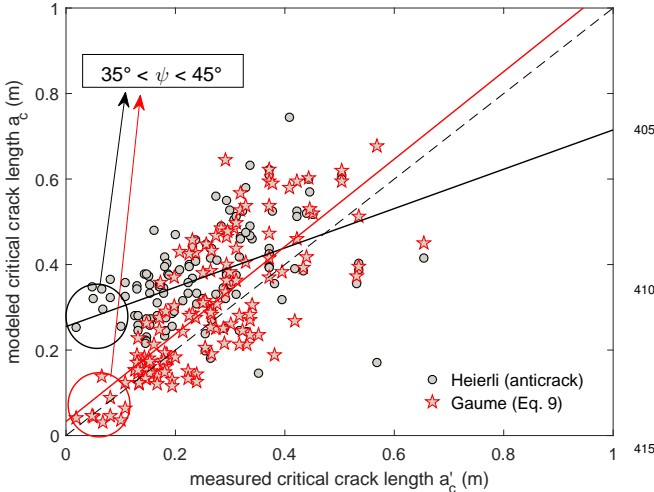

Figure 6: Comparison between measured and modeled critical crack lengths using the anticrack model (Heierli et al., 2008) (black circles) and our new model (Eq. 9, red stars). The continuous lines represent linear fits; in black: $a_c = \gamma_H a'_c + \delta_H$ with $\gamma_H = 0.46$ (0.32, 0.61), $\delta_H = 0.23$ (0.21, 0.30) and $R_H^2 = 0.24$; in red: $a_c = \gamma_G a'_c + \delta_G$ with $\gamma_G = 1.023$ (0.872, 1.173), $\delta_G = 0.03$ (-0.01, 0.07) and $R_G^2 = 0.58$. The numbers in brackets represent the 95% confidence interval. The dashed line represents the 1:1 line.

For low-angle terrain, the anticrack model and our new formulation yield similar results. However, this is where the similarities end. Indeed, overall the anticrack model overestimates $a_c$ and more closely resembles a model which only accounts for stresses due to slab bending: $a_c^b = \Lambda \sqrt{\tau_p / \sigma_n}$ (obtained by solving $\tau_{max}^b = \tau_p$). For steep slopes ($\psi > 30°$), where the shear stress at the crack tip due to slab bending becomes negligible compared to that due to slab tension, critical crack length values obtained from Eq. 9 strongly differ from the prediction of the anticrack model and converge on the contrary towards a purely tensile model, generally referred to as "pure shear model": $a_c^t = \Lambda(\tau_p / \tau_g - 1)$ (obtained by solving $\tau_{max}^t = \tau_p$, Fig. 4e).

Heierli et al. (2008) illustrated the low dependence of the critical crack length on slope angle with results from field experiments presented by Gauthier and Jamieson (2008). However, these PST experiments were performed on a non-persistent WL consisting of precipitation particles and measurements made on the flat were performed one day before the experiments made on slopes (Gauthier, 2007). This indicates that the trend with slope angle may be influenced by the burial time of the WL since sintering and settlement effects can strongly affect snowpack properties within one day, especially with the layer of precipitation particles which was tested (Szabo and Schneebeli, 2007; van Herwijnen and Miller, 2013; Podolskiy et al., 2014). Furthermore, Heierli et al. (2008) assumed snow cover properties independent of

slope angle which is somewhat questionable, since snowpack properties can also change with slope angle, thus obscuring the true slope angle influence. As an example, for their validation, Heierli et al. (2008) assumed a constant slab thickness $D = 11$ cm over the different slope angles $\psi$, while $D$ decreased with increasing $\psi$ according to Gauthier and Jamieson (2008). In addition, it is also known that weak layer strength (Reiweger et al., 2015), slab density (Endo et al., 1998) and thus the elastic modulus (Scapozza, 2004) are strongly depend on slope angle. Hence we argue that the dependence of the critical crack length on slope angle obtained from a model with fixed value of the other parameters should not be compared to the trend observed in the experiments which is the result of a combination of many varying properties. Instead, one should directly compare the measured critical crack length to the modeled one, taking as input parameters the properties measured at the location where the PST was performed.

By comparing the anticrack model to the 93 PST measurements (Fig. 6), we see that $a_c$ is generally overestimated, especially for short critical crack lengths and steep slopes ($35° < \psi < 45°$). For higher values of $a_c$ and gentler slopes, the anticrack predictions better agree with our formulation, even though they still remain mostly above the 1:1 line.

**Slope angle dependence**

We showed that the critical crack length $a_c$ decreases with increasing slope angle $\psi$ for a PST with slope-normal faces, a constant slab thickness $D$ and constant values of the mechanical properties. However, the rate of decrease of $a_c$ with $\psi$ is strongly influenced by the elastic modulus $E$ and thickness $D$ of the slab. Low values of $E$ and/or $D$ lead to a gentler decrease of $a_c$ with $\psi$ (Fig. 7).

Yet, if slab depth $H$ (vertical) is constant with respect to slope angle, the slab thickness decreases with increasing slope angle according to $D = H \cos \psi$. Since a lower slab thickness leads to a higher critical crack length (Fig. 4c) this effect leads to an apparent reduction of the decrease of $a_c$ with $\psi$. As an illustration, we compare our model (Eq. 9) to the PST experiments presented Bair et al. (2012) for which the slab density and elastic modulus were very low (storm snow, $\rho = 84$ kg/m³, $E = 0.22$ MPa; Fig. 8). The low elastic modulus thus leads (Eq. 9) to a very gentle decrease of $a_c$ with $\psi$ in line with the experimental data. The anticrack model was also plotted in Fig. 8a and shows very comparable results. Yet, the values of the WL specific fracture energy $w_f$ and slab elastic modulus $E$ in Bair et al. (2012) were estimated by a fit of the anticrack model to the data using the method described by van Herwijnen et al. (2010, 2016) which explains the good agreement. Interestingly, also a pure shear model (Eq. 7) with the same input parameters as for our model (Eq. 9) would lead to a reasonable agreement for steep slopes ($\psi > 30°$). In the studies of Heierli et al. (2008) and Bair et al. (2012), the significant difference obtained be-

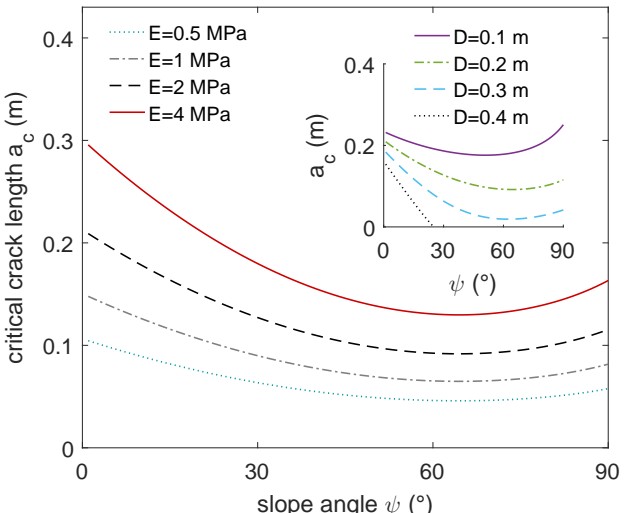

Figure 7: Effect of the slab elastic modulus on the slope angle dependency of the critical crack length (Eq. 9) for $\rho = 200\,\mathrm{kg/m^3}$, $D = 0.2\,\mathrm{m}$, $D_{wl} = 4\,\mathrm{cm}$. Inset: Effect of slab thickness on the slope angle dependency of the critical crack length for $\rho = 200\,\mathrm{kg/m^3}$, $E = 2\,\mathrm{MPa}$, $D_{wl} = 4\,\mathrm{cm}$.

tween the anticrack model and the pure shear model (Mc-Clung, 1979; Gaume et al., 2013) is an artifact simply due to the fact that the same specific fracture energy was taken as input for both models although the underlying physical assumptions are strictly incompatible: the pure shear model considers a quasi-brittle behavior for the weak layer and the anticrack model considers a purely ridid behavior. In fact, for $\psi > 30°$ and short critical crack lengths which are typically encountered in field experiments, Gaume et al. (2014b) recently showed from the energy balance equations that both approaches lead to very comparable results, which is confirmed by our new model.

Finally, geometrical effects significantly influence how the critical crack length depends on slope angle. Figure 8b shows the critical crack length as a function of slope angle for three different PST configurations: (i) constant slab thickness $D$ and slope normal faces (SNF); constant slab depth $H$ and and slope normal faces (SNF); (iii) constant slab depth and vertical faces (VF). The vertical character can be accounted for by adding $D/2\tan\psi$ to the critical crack length as proposed by Heierli et al. (2008) (see supplement). We clearly observe that the decrease of $a_c$ with $\psi$ is gentler with a constant slab depth $H$ than with a constant slab thickness $D$ as shown before. In addition, we observe an increase of the critical crack length with increasing slope angle if the PST is made with vertical faces and if the slab depth is constant. This is in line with the PST experiments of Gauthier and Jamieson (2008) performed with vertical faces and a constant slab depth $H$. It seems that Heierli et al. (2008) did account neither for the vertical character of the faces nor for the decrease of slab

thickness with slope angle in their comparison to the data of Gauthier and Jamieson (2008). The increasing trend predicted by our model with a constant slab depth $H$ and vertical faces might explain why the Extended Column Test (ECT) scores were observed to increase with increasing slope angle (Heierli et al., 2011; Bair et al., 2012).

**Relevance and limitations**

Performing DEM simulations allowed us to investigate crack propagation in weak snow layers without relying on the same strong assumptions concerning the weak layer as previous research (McClung, 1979; Chiaia et al., 2008; Heierli et al., 2008; Gaume et al., 2014b). For the sake of developing theoretical models, these studies considered either a purely interfacial weak layer (McClung, 1979; Chiaia et al., 2008; Gaume et al., 2014b) or a weak layer composed of a completely rigid material with a slope-independent failure criterion (Heierli et al., 2008). On the contrary, in our simulations, the weak layer is characterized by a finite thickness, an elasticity and a mixed-mode failure envelope in line with results of recent laboratory experiments (Reiweger et al., 2015). These DEM simulations can thus be seen as numerical laboratory experiments in which the effect of slab and weak layer properties on crack propagation can be investigated independently (which is impossible to do in the field) and from which analytical expressions can be inferred using a strength-of-material approach. This important step forward allows to reconcile the shear- and collapse-based approaches. For example, our model can describe crack propagation in flat terrain providing the same results as the anticrack model. Furthermore, it predicts the decrease of the critical crack length with increasing slope angle in line with shear-based models (McClung, 1979; Chiaia et al., 2008; Gaume et al., 2014b) and in contrast with the anticrack model since the latter assumes rigidity and slope-independent failure of the weak layer. Note that in the simulations and in reality, slab bending also induces shear stresses within the slab leading to possible slope normal stress variations in the WL. This effect is not accounted for in our analysis. However, the good agreement between Eq. 9 and DEM results (Fig. 4) suggests that it is in fact of second order, thereby validating the assumption that the maximum shear stress at the crack tip has two main contributions related to slab tension and bending (Eq. 6).

In a recent study, Gaume et al. (2015b) showed that the DEM model can also reproduce the dynamic phase of crack propagation as well as fracture arrest in the slab which was treated as an elastic-brittle material. In particular, the crack propagation speed and distances obtained by PTV analysis of the PST were well reproduced. It was also shown that the propagation distance (distance between the lower edge and slab fracture) was almost always higher than the critical crack length except for combinations of very low slab densities and thicknesses. This behavior is also observed in field

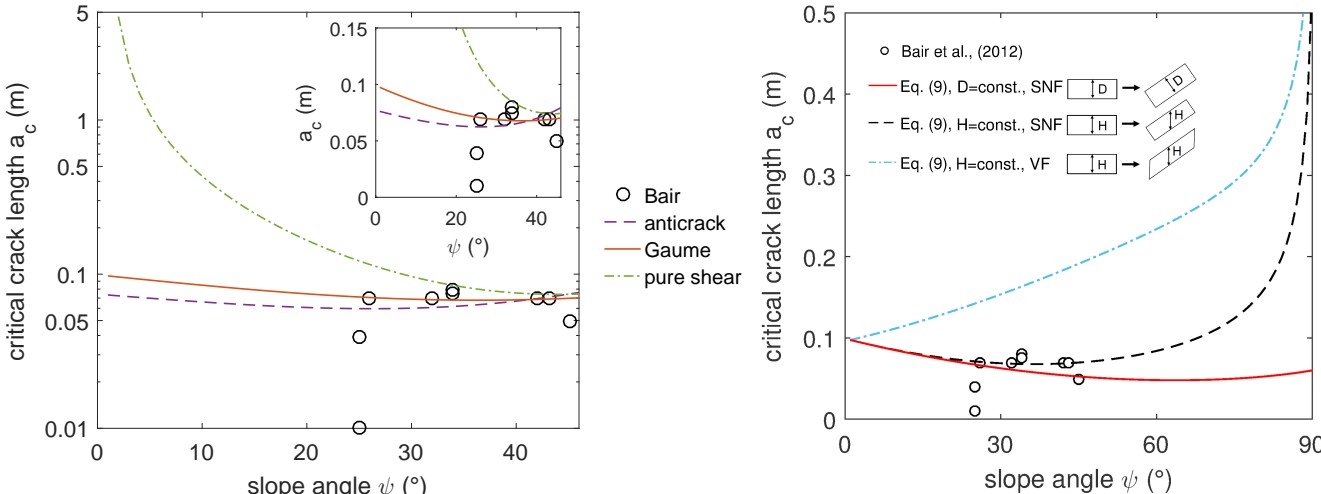

Figure 8: (a) Critical crack length vs slope angle: comparison between the data of Bair et al. (2012) (black circles) and our new model (Eq. 9, red line), the anticrack model (purple dashed-line) and a pure shear model (Eq. 7, green dotted line) for a constant slab depth $H = 0.35\,\mathrm{m}$ ($D = H \cos\psi$) and the same input parameters as in Bair et al. (2012) with a semi-log scale. The cohesion $c = 500\,\mathrm{Pa}$ was estimated based on the hand hardness index provided in Bair et al. (2012) using the parametrization by Geldsetzer and Jamieson (2001) and Jamieson and Johnston (2001). Inset: linear scale. (b) Effect of geometry on the slope angle dependency for the PST. SNF: Slope normal faces. VF: Vertical faces. const.: constant.

experiments. Accordingly, treating the slab as a linear elastic material before the onset of crack propagation is justified. This assumption was also confirmed by recent field studies (van Herwijnen et al., 2010, 2016) showing that the slab displacement obtained with particle tracking can be described by beam theory with a linear elastic assumption. Hence, with the present study, we show that our DEM model is able to address the whole crack propagation process.

The main limitation of our model is the uniform character of the slab. In this paper, the multilayered character of the slab was not accounted for, for clarity reasons since the phenomenon is already very complex. However, the elastic moduli of the slab layers have a very important influence on slab deformation and thus on the critical crack length (Reuter et al., 2015). For the comparison with the experiments, the elastic modulus was computed from the average slab density. However, in practice, a slab with a uniform density $\rho$ will deform differently than a slab of average density $\rho$ consisting of various layers with contrasting properties. This is probably the reason why significant scattering is observed in Fig. 6 although the overall agreement is good.

Concerning the weak layer, the schematic microstructure considered in this study is sufficient to capture the main features of the failure envelope (Chandel et al., 2014). Considering more complex microstructures for the weak layer might lead to a better quantitative agreement with experimental data. For instance, it was shown (Gaume et al., 2014a) that with a weak layer produced by ballistic deposition, the experimental failure envelope obtained by Reiweger et al.

(2015) could be reproduced. In the future, performing numerical simulations accounting for the real microstructure of weak snow layers, as derived from X-ray microtomographic images (Hagenmuller et al., 2014), represents an interesting prospect. Nevertheless, if such refinements can certainly have an impact on the shear strength value $\tau_p$, they should not change the structure of Eq. 9.

Another important aspect is the relevance of our new model with regards to slab avalanche release. We showed that our model was able to reproduce crack propagation at the scale of the PST. However, at the slope scale, 3D effects, slope-transverse propagation, terrain and snowpack variability (Schweizer et al., 2008; Gaume et al., 2015a) might make the process even more complex. Nevertheless, it was shown that the critical crack length correlates very well with signs of instability (Reuter et al., 2015). In particular, they showed that no signs of instability were recorded for $a_c > 0.4$ m while whumpfs, cracks and avalanches were observed for $a_c < 0.4$ m. Hence, our new model of critical crack length can be of major importance in view of avalanche forecasting.

**Application to simulated snow stratigraphy**

The snow cover model SNOWPACK (Bartelt and Lehning, 2002; Lehning et al., 2002a, b), which simulates the temporal evolution of snow stratigraphy, is used for operational avalanche forecasting in Switzerland. Potential weak layers in the simulated snow profiles are identified by calculating the structural stability index (SSI), an index based on the balance between shear stress and shear strength (Schweizer

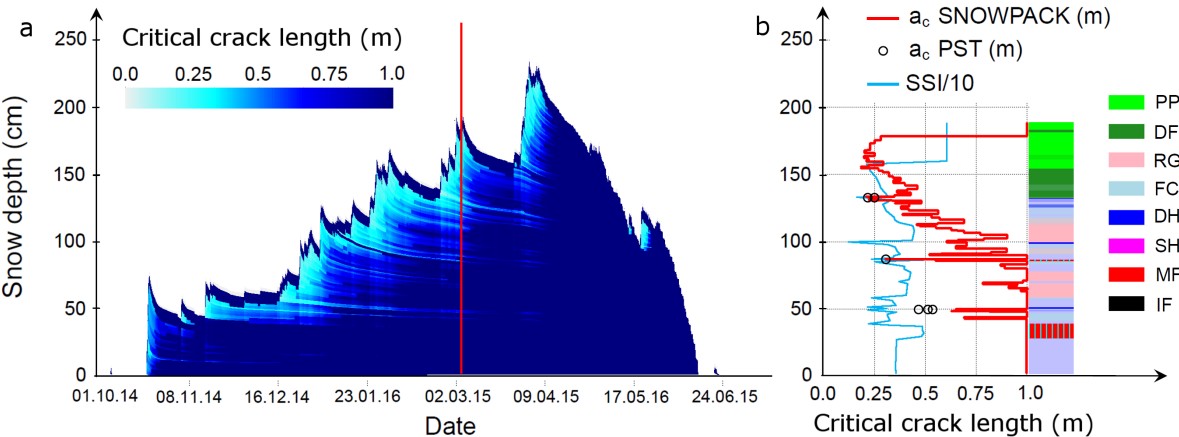

Figure 9: (a) Seasonal profile of the simulated critical crack length (winter 2014-2015) at Steintälli (Davos, Switzerland) on the flat. (b) Vertical profile of the critical crack length (modeled and from field PSTs) and SSI/10 for the date marked by the vertical red line in (a). The grain type is shown on the right following Fierz et al. (2009).

et al., 2006; Monti et al., 2012). The SNOWPACK model also provides all necessary variables to determine the critical crack length based on Eq. 9. To demonstrate the practical applicability, we performed a simulation for the 2014-2015 winter at the location of an automatic weather station above Davos, Switzerland (Fig. 9). Note that the critical length was arbitrarily set to 1 m in the first 10 cm, since avalanche probability for such shallow layers is generally very low (van Herwijnen and Jamieson, 2007). The same was done when computed values of the critical length exceeded 1 m. Short critical crack lengths clearly highlight potential WLs in the snowpack during the season (Fig. 9a). At the end of the dry-snow season, around 10 April, the percolation of liquid water into the snow cover resulted in a rapid increase in shear strength and thus in larger critical crack lengths throughout the snow cover.

On 3 March 2015 we performed several PSTs on three WLs at the location of the automatic weather station. The SNOWPACK simulation for that specific day clearly shows local minima in the calculated critical crack length for these three WLs (Fig. 7b). Modeled critical crack lengths were in good agreement with PST field measurements (black circles in Fig. 9b), and SNOWPACK was able to reproduce the observed increase in $a_c$ with increasing depth of the WL. Schweizer et al. (2016) recently followed the temporal evolution of the critical cut length and showed that the implementation of Eq. 9 is very sensitive to the parametrization of $\tau_p$ used in SNOWPACK (Jamieson and Johnston, 2001; Schweizer et al., 2006). Finally, layers for which critical crack lengths were lower generally also corresponded to layers with local minima in the SSI, suggesting that a combination of SSI and $a_c$ may provide a more reliable instability criterion (Reuter et al., 2015).

## 5   Conclusions

We proposed a new analytical expression to assess the conditions for the onset of crack propagation in weak snowpack layers. The formulation was developed based on discrete element simulations; it accounts for crucial physical processes involved in crack propagation in snow, namely the complex mechanical behaviour of the WL and the mixed stress states in the slab induced by slab tension and bending resulting from WL collapse. A critical parameter in the formulation is the lengthscale $\Lambda$, which accounts for the elastic mismatch between the slab and the WL.

The analytical expression for the critical crack length reproduced field data obtained with 93 propagation saw test experiments. In contrast, the anticrack model (Heierli et al., 2008) which, although appropriate for flat terrain, significantly overestimated the critical length for steep slopes, where avalanches release. Furthermore, our model predicts that the critical crack length decreases with increasing slope angle. This shows that triggering an initial failure leading to slab avalanche release is more likely on steep rather than on low-angle slopes, a rather intuitive result. Nevertheless, our model still allows for crack propagation on flat terrain and remote triggering of avalanches, both of which are widely documented by countless field observations.

Finally, our new expression was implemented in the snow cover model SNOWPACK to evaluate the critical crack length for all snow layers throughout the entire season. While validation is still required, this opens promising perspectives to improve avalanche forecasting by combining traditional stability indices with a new metric to evaluate crack propagation propensity.

*Acknowledgements.* The critical crack length is implemented in the SNOWPACK model, which is available under the GNU Lesser General Public Licence Version 3 and can be retrieved at http://models.slf.ch. We are grateful to all SLF colleagues who assisted in field data collection. We thank Benjamin Reuter for the SMP-derived specific fracture energy data and for insightful discussions and comments on the paper. We acknowledge the constructive comments of two anonymous reviewers as well as Ned Bair who helped us to improve our paper. Johan Gaume has been supported by the Ambizione grant of the Swiss National Science Foundation (PZ00P2_161329).

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
