# Peer review of "Date: 5 December 2016"

_The Cryosphere, 2016_

## Referee Comment (RC1) · Anonymous Referee #1 · 29 Apr 2016

Overview This manuscript addresses a topic that is relevant to snow and avalanche mechanics. It addresses a topic of current interest and debate. Specifically, a model for a layered snow cover consisting of a "weak layer" with a flaw or crack that is overlain with a homogeneous elastic "slab" layer is presented. It addresses conditions for the the critical crack length within the weak layer that will result in its failure and propagation. The primary interest in the topic is the consequence of failure with respect to avalanche initiation. The paper builds on previous work and suitably acknowledges those contributions. Several, but not all, of the earlier contributions utilized a linear elastic fracture mechanics approach. The approach presented in this manuscript uses a discrete element method (DEM), conceptually based on a mechanics of materials or elasticity approach, incorporating the concept of stress concentrations leading to propagation. From this an analytic expression is developed. Perhaps the major contribution of the

paper is to add to the discussion the influence of slope angle on weak layer failure. This is in contradiction to a widely accepted notion that the fracture is slope independent. It also attempts to add a more robust inclusion of the independent material properties of the layers. The results are intuitively reasonable. That said I have a number of questions and comments that I feel need to be addressed. A number of these comments are suggested in order more precisely clarify details to relate the physical description to what is calculated in the model. This is important, especially since it provides results that are counter to earlier work. There are some more technical issues that need to be explained or justified. I feel that this is an interesting and relevant paper that I would like to see published if the comments can be adequately addressed. I am presuming that the the issues can readily be resolved and clarified. Assuming that this is the case, the revisions should be a relatively minor effort by the authors. However, I am rating this version as major revision since I feel that the paper needs to be reviewed again prior to acceptance. Detailed comments Line 27 I don't think you mean to say that cracks form below an overload, but that the additional overload in association with a crack may lead to failure. Line 63 The specific fracture energy would have to be between the slab and the rigid weak layer, since the rigid material cannot support elastic potential energy. I think that this term is an ill defined in the original paper. Line 51 suggest ...which {allowed to solve} the problem... to ...which {allowed solution} to the problem... Line 80 – suggest ...anticrack model, {these} strength of material... The strength of materials approach certainly can account for bending. I suggest that you make the change to emphasize that the methods implemented for snow that you reference do not have bending. Line 98 Does soft contact imply that the contacts (bonds) are elastic. Are the grains taken to be rigid in your simulations? Figure 2 You should present labels for ac, $\lambda$ and Io in caption. Caption Fig 2 $\tau g = gD \sin(\Psi)$ This would be the shear stress at the interface of slab and WL. Thickness of WL is not included. Does this then imply that the stress concentration is at the interface between the slab and WL or is shear stress in WL assumed uniform throughout the thickness? Caption Fig 2 How is the residual stress explicitly defined? Is this residual stress used anywhere in this development?

Line 113 How does this compare with Scapozza, equation 5? Both the slab and WL are taken to be isotropic. The triangular structure may, perhaps in a future iteration, allow for anisotropy for forms such as surface hoar and depth hoar. Figure 3 Perhaps call normal stress the "slope normal stress". The normal stress value, as well as shear, is a function of the coordinate system chosen. Caption You should state that compression is taken as positive. Caption $\Psi$ is labeled in the text as the slope angle. Should this be a critical slope angle? Should this be $\Psi t$ instead? Line 115 I'm not clear why you can state that these are satisfactory based on fig 3? While the initial slopes of increasing normal stress are similar, the actual values are quite dissimilar. Normal stresses are actually of opposite sign for the initial slope on the figure? Line 125 "specific fracture energy" is used only in the anticrack model. Correct? "...the penetration resistance using the snow micropenetrometer (SMP) in the weak layer according to Reuter et al." This would not be possible if the layer was actually rigid, as is assumed for the anticrack model. Resistance would be infinite. Eq 1 $\Psi t$ is not defined. Line 140 So is K = 1 kPa ? From figure Eq 5 – Is this for the slab only? Or both slab and WL? Line 148 – Is this Poisson's ratio for the slab and the weak layer? Is this used for calculation of the elastic mismatch? Is the elastic mismatch used in your development or merely mentioned as an aside? Line 151 I presume that you mean to relate the "tension in the slab" to imply the slope parallel normal stress due to gravity. Slab bending may also cause a slope parallel tension component as well compression and shear in this same coordinate orientation. Line 152 Is the crack tip calculated to be the thickness of the weak layer or at the bonds between the slab and WL? If it is at the interface, as appears to be implied at some points in the presentation, it would seem that this may differ from the max within the WL. Figure 4 Young's Modulus, E, is a function of density, in equation 5. Figure 4a has the critical crack varying with E with a fixed density of 300 kg m3. Similarly, figure 4b has a fixed E with crack length varying with density. Something seems wrong here? Is this somehow related to the calculation of E described in line 112? Line 163 Is this statement implying that you are considering only the influence of shear stress on the stress concentration? Relating to failure envelope? Again, is this assumed to be at the

interface between the layers or within the WL itself? Equation 7 You are examining the case in which "WL has no thickness". But at this point $\lambda$ (line 171) goes to zero so a/$\lambda$ goes to infinity. The max shear in the WL then goes to infinity in equation 7? There is a singularity at the crack tip. This is also apparent in the bending component in equation 8. Even for any very thin layer of finite thickness this will be huge. This needs to be discussed and explained. Line 177 – suggest {slope} normal stress. Line 180 – If you assume a rigid weak layer, the elastic properties of the weak layer are irrelevant. It is also not clear why the rigid layer is assumed if the shear stress is independent of these properties. Line 180 I think you need to more explicitly describe the beam boundary conditions assumed. "…according to beam theory" is rather vague. I am also not seeing where this reference (Timoshenko and Goodier, 1970), is dealing with cracks, except for a brief mention relative to Griffith fracture. Can you be more specific? Line 182 – suggest replace {allows} with {provides a means} Line 182 Is it using Dundur's elastic mismatch parameters in the numerical solution? Does this imply that you are calculating the crack to be at the interface between layers? How does scaling with $\lambda$ instead of D come into play? Line 191 I presume that the residual stress, $\tau r$, is not used as the result of this condition. And this paper is not concerned with the sliding following failure of the week layer. Line 211 This is an important result, since in the avalanche community the concept of slope independence has been quite widely accepted of late. Line 215 Here again I think that it should be clearly stated that you are referring to the stress due to the gravity along the slope and independently, the "stress state" as the result of bending. Line 215 Parts of this development seem to imply that the failure would be at the bonds between the slab and the weak layer. I think that what is being calculated should be clearly stated. As an aside, in the physical situation for depth hoar the failure may well be at the lower interface or at the upper and lower. The crystals themselves often appear to remain intact. Line 221 – suggest changing "…accounts for slab bending only…" to "accounts for {stresses due to} slab bending only" Line 223 Should clearly state that you are referring to the shear stress at the crack tip induced by the slope parallel loading of the slab. This results in the shear stress at the interface

between the slab and WL. Line 224 ac t goes toward zero as WL thickness goes to zero. Following discussion above. Figure 6 caption – suggest changing "represents" to "represent" Line 227 – suggest changing "decreases" to "decrease". Line 252 – I think that you should state that you are using a mechanics of materials or perhaps an elasticity approach, if that is appropriate. Line 254 Suggest changing "...allows to reconcile shear and collapse..." to "...reconciles the shear and collapse..." Line 321 – suggest changing ...that {skier triggered avalanches are} more likely on steep rather than on flat slopes..." to "...that {avalanche initiation} is more likely on steep rather than on flat slopes..."

[Figure]

Overview

This manuscript addresses a topic that is relevant to snow and avalanche mechanics. It addresses a topic of current interest and debate. Specifically, a model for a layered snow cover consisting of a "weak layer" with a flaw or crack that is overlain with a homogeneous elastic "slab" layer is presented. It addresses conditions for the the critical crack length within the weak layer that will result in its failure and propagation. The primary interest in the topic is the consequence of failure with respect to avalanche initiation. The paper builds on previous work and suitably acknowledges those contributions. Several, but not all, of the earlier contributions utilized a linear elastic fracture mechanics approach. The approach presented in this manuscript uses a discrete element method (DEM), conceptually based on a mechanics of materials or elasticity approach, incorporating the concept of stress concentrations leading to propagation. From this an analytic expression is developed.

Perhaps the major contribution of the paper is to add to the discussion the influence of slope angle on weak layer failure. This is in contradiction to a widely accepted notion that the fracture is slope independent. It also attempts to add a more robust inclusion of the independent material properties of the layers. The results are intuitively reasonable.

That said I have a number of questions and comments that I feel need to be addressed. A number of these comments are suggested in order more precisely clarify details to relate the physical description to what is calculated in the model. This is important, especially since it provides results that are counter to earlier work. There are some more technical issues that need to be explained or justified.

I feel that this is an interesting and relevant paper that I would like to see published if the comments can be adequately addressed. I am presuming that the the issues can readily be resolved and clarified. Assuming that this is the case, the revisions should be a relatively minor effort by the authors. However, I am rating this version as major revision since I feel that the paper needs to be reviewed again prior to acceptance.

Detailed comments

Line 27   I don't think you mean to say that cracks form below an overload, but that the additional overload in association with a crack may lead to failure.

Line 63   The specific fracture energy would have to be between the slab and the rigid weak layer, since the rigid material cannot support elastic potential energy. I think that this term is an ill defined in the original paper.

Line 51 suggest ...which {allowed to solve} the problem... to ...which {allowed solution} to the problem...

Line 80 – suggest ...anticrack model, {these} strength of material...
The strength of materials approach certainly can account for bending. I suggest that you make the change to emphasize that the methods implemented for snow that you reference do not have bending.

Line 98   Does soft contact imply that the contacts (bonds) are elastic. Are the grains taken to be rigid in your simulations?

Figure 2   You should present labels for $a_c$, $\lambda$ and $l_o$ in caption.

Caption Fig 2   $\tau_g = \rho g D \sin(\Psi)$   This would be the shear stress at the interface of slab and WL. Thickness of WL is not included. Does this then imply that the stress concentration is at

**Fig. 1.**

---

## Referee Comment (RC3) · Anonymous Referee #2 · 17 May 2016

In the submitted manuscript by Gaume et al. "Snow fracture in relation to slab avalanche release: critical state for the onset of crack propagation", a Discrete Element Model (DEM) of Propagation Saw Tests (PSTs) is used to examine the effects of varying material properties and other model parameters. Analytical expressions using a Mohr-Coloumb Cap model are derived and compared to model results. An analytical expression for the critical length is derived and compared with field PSTs along with expressions from other models. I applaud the authors for their efforts, as this is an important area of snow avalanche research with problems that have not been resolved, however I suggest that this manuscript requires major revisions to become publishable.

I am not a DEM expert, but the model set up and results appear sound to me. The analytical work is also sound. My main criticism is that the analytical model does not explain previous fieldwork with PSTs [Gauthier and Jamieson, 2008; McClung,

2009; Bair et al., 2012] showing that snowpacks with identical slab thickness (slope normal) and densities showed the same or increasing critical cut lengths as slope angle increased. Note that the same behavior has been shown in Extended Column Tests [Heierli et al., 2011; Bair et al., 2012] and Compression Tests [Birkeland et al., 2014], all with the same slab thickness and density. I realize those tests were not modelled here, but the fact that the three main stability tests all show the same trend cannot be ignored. Rather than addressing this discrepancy directly, the authors dismiss previous fieldwork with PSTs by stating that the snowpack properties could not have remained the same throughout the range of slope angles tested, thereby calling into question the accuracy of the reported values in each of these studies. As the author of one of these previous field studies, I find this particularly irksome given the trouble that I went to in ensuring uniform conditions throughout the slope angles tested. Further, the authors do not present any field evidence to support their modeled slope angle dependence, i.e. PSTs showing decreasing critical length with increasing slope angle given constant slab thickness and density. Finding constant slab thickness and density over a range of slope angles in the field is not that difficult, which explains why multiple studies have been able to do it. Perhaps the authors need to re-read these studies more carefully. For instance, they state that Heierli et al. [2008] only used three PSTs to validate the anticrack model. In fact, it was 44 PSTs over three different slope angles [Gauthier and Jamieson, 2008] used, with the means taken for each of the three slope angles. Likewise, one of the major criticisms of the anticrack model has been that it assumes a linear elastic slab. This model is supposed to be an improvement in accuracy, but the authors make the same linear elastic assumption without discussing pitfalls. This may explain some of the scatter in Figure 6 for both models. Further, the evaluation of the anticrack model is flawed since a constant (0.1 J m-2) specific fracture energy is used. Field data show this value varies by more than an order of magnitude [Schweizer et al., 2011]. A constant specific fracture energy in the anticrack model is akin to setting a constant value for the weak layer shear strength in the authors' model.

The manuscript requires extensive editing to correct grammatical errors. I suggest an

English language service. For instance, there were too many errors of tense for me to correct. I gave up. Likewise, symbols are used in graphs and equations without being defined until later in the text. I suggest deleting the "Application to simulated snow stratigraphy" section, as it is not convincing with only one measured and modelled profile for comparison. Other minor corrections are included as an annotated PDF.

Bair, E. H., R. Simenhois, K. Birkeland, and J. Dozier (2012), A field study on failure of storm snow slab avalanches, Cold Regions Science and Technology, 79-80, 20-28, doi: 10.1016/j.coldregions.2012.02.007. Birkeland, K. W., E. H. Bair, and D. Chabot (2014), The effect of changing slope angle on compression test results, in International Snow Science Workshop, edited, pp. 746-751, Banff, Canada. Gauthier, D., and B. Jamieson (2008), Evaluation of a prototype field test for fracture and failure propagation propensity in weak snowpack layers, Cold Regions Science and Technology, 51(2-3), 87-97, doi: 10.1016/j.coldregions.2007.04.005. Heierli, J., P. Gumbsch, and M. Zaiser (2008), Anticrack nucleation as triggering mechanism for snow slab avalanches, Science, 321(5886), 240-243, doi: 10.1126/science.1153948. Heierli, J., K. W. Birkeland, R. Simenhois, and P. Gumbsch (2011), Anticrack model for skier triggering of slab avalanches, Cold Regions Science and Technology, 65(3), 372-381, doi: 10.1016/j.coldregions.2010.10.008. McClung, D. M. (2009), Dry snow slab quasi-brittle fracture initiation and verification from field tests, Journal of Geophysical Research, 114, F01022, doi: 10.1029/2007JF000913. Schweizer, J., A. van Herwijnen, and B. Reuter (2011), Measurements of weak layer fracture energy, Cold Regions Science and Technology, doi: 10.1016/j.coldregions.2011.06.004.

Please also note the supplement to this comment:
http://www.the-cryosphere-discuss.net/tc-2016-64/tc-2016-64-RC3-supplement.pdf

———————————————————

---

## Author Comment (AC1) · 7 Jul 2016

We greatly appreciate the relevant discussion raised by the reviewers' comments that will contribute to improve our paper and also to argue and discuss a very hot topic in the snow community. The comments of Reviewer #1 concern clarifications of the model and assumptions and Reviewer #2 raised the issue of the slope angle dependency in field experiments. We acknowledge the complementary nature of these two reviews and we believe that the open-access character of this discussion will also represent an important addition to the paper itself. We provide detailed answers to their concerns below and we will revise our paper accordingly.

**Reply to Referee #1**

We thank Referee #1 for his constructive comments that helped us to significantly improve the quality of our paper and clarify some points of the model that might have been unclear.

We provide below detailed answers to the various issues raised by the reviewer and we will revise our paper in order to account for his/her remarks.

**Comment M1).** *Line 27: I don't think you mean to say that cracks form below an overload, but that the additional overload in association with a crack may lead to failure*

**Answer to comment M1).** In fact, we mean it as we write it. At the depth of the weak layer, if the additional stress due to an overload exceeds the weak layer strength, the weak layer fails locally but it does not mean that an avalanche would release. The size of this local failed zone - what we call crack - needs also to exceed the critical crack length. Our point here was to explain where cracks come from in the snowpack. Snow is inherently full of pre-existing flaws and cracks at the microscale but "macroscopic" cracks like those artificially made with a snow saw in the PST, in our opinion, should form in super weak zones of the snowpack or below local overloads.

**Comment M2).** *Line 63 The specific fracture energy would have to be between the slab and the rigid weak layer, since the rigid material cannot support elastic potential energy. I think that this term is an ill defined in the original paper*

**Answer to comment M2).** The specific fracture energy in the anticrack model is indeed for the interface between the slab and the weak layer, assumed rigid for the purpose of deriving the mechanical energy of the slab. We agree it was not clearly mentioned in the original paper of Heierli et al. (2008) but was precisely described in the thesis of Dr. Heierli. It is generally referred to as the "weak layer specific fracture energy" for simplicity reasons but sometimes lead to confusions. This property characterizes the energetic cost of creating new fracture surfaces. It should not be confused with the potential strain energy, which is related to the behavior of the material, and is directly proportional to fracture toughness for an elastic material.

**Comment M3).** *Line 51 suggest ...which allowed to solve the problem... to ...which allowed solution to the problem...*

**Answer to comment M3).** This will be changed as suggested.

**Comment M4).** *Line 80 suggest ...anticrack model, these strength of material...*

**Answer to comment M4).** This will be changed as suggested.

**Comment M5).** *The strength of materials approach certainly can account for bending. I suggest that you make the change to emphasize that the methods implemented for snow that you reference do not have bending.*

**Answer to comment M5).** We agree. This will be changed as suggested.

**Comment M6).** *Line 98 Does soft contact imply that the contacts (bonds) are elastic. Are the grains taken to be rigid in your simulations?*

**Answer to comment M6).** Both grains and bonds are elastic and the bonds have a brittle failure criterion (in shear and tension). We use as contact law the parallel bond model of PFC2D (elastic-brittle). This contact law is fully described in our recent paper Gaume et al. (2015). We believe it is enough here to refer to our paper but we will modify the last sentence of the paragraph as follow: "The numerical setup and the cohesive contact law implemented is fully described in Gaume et al. (2015).".

**Comment M7).** *Figure 2 You should present labels for ac, λ and lo in caption.*
**Answer to comment M7).** This will be done.

**Comment M8).** *Caption Fig 2 $\tau_g = \rho g D \sin(\psi)$ This would be the shear stress at the interface of slab and WL. Thickness of WL is not included. Does this then imply that the stress concentration is at the interface between the slab and WL or is shear stress in WL assumed uniform throughout the thickness?*
**Answer to comment M8).** Weak layers are usually significantly thinner than the slab so $D >> D_{wl}$. In addition, the density of the weak layer, in particular with the very porous triangular structure used, is low compared to slab density so the contribution of the weak layer to the overall load is negligible. We thus assume that $\tau_g = \rho g D \sin(\psi)$ inside the weak layer, regardless of the vertical position. This assumption is usually made in the avalanche community. We did not mention this point but we will thus clarify it by adding the following sentence in section 3 - DEM Simulations. "We assume the shear stress inside the WL to be due to the slab weight only $\tau_g = \rho g D \sin \psi$ (WL weight neglected)."

**Comment M9).** *Caption Fig 2 How is the residual stress explicitly defined? Is this residual stress used anywhere in this development?*
**Answer to comment M9).** The residual stress $\tau_r$ between the slab and the remaining part of the weak layer is defined as follows: $\tau_r = \sigma_n \tan \phi = \rho g D \cos \psi \tan \phi$ where $\phi$ is the friction angle. However, this residual stress is not used in the current development because crack propagation occurs always before reaching the length $l_0$ i.e. before the slab touches the broken weak layer. If it was not the case, one should correct $\tau_{max}$ which would then be equal to:

$$\tau_{max} = \tau_g \left(1 + \frac{l_0}{\Lambda}\right) + \frac{1}{2}\sigma_n \left(\frac{l_0}{\Lambda}\right)^2 - \tau_r \left(1 + \frac{a - l_0}{\Lambda}\right).$$

In this case, the critical crack length would have a different expression.

**Comment M10).** *Line 113 How does this compare with Scapozza, equation 5?*
**Answer to comment M10).** The statement line 113 is not related to Scapozza at all. We describe how we back-calculate the contact (micro) elastic properties from the macroscopic (bulk) elastic modulus $E$. This procedure is fully described in Gaume et al. (2015) and was shortly recalled here. We introduce Scapozza's relationship between density and the elastic modulus in a section called "Comparison with PST experiments" and so we only use it for the comparison with field data or in the implementation of Eq. 9 into SNOWPACK.

**Comment M11).** *Both the slab and WL are taken to be isotropic. The triangular structure may, perhaps in a future iteration, allow for anisotropy for forms such as surface hoar and depth hoar.*
**Answer to comment M11).** In fact, the slab is isotropic due to the regular granular lattice used to model it. On the other hand, the triangular structure of the weak layer very likely leads to a non-isotropic behavior. However, in our case, speaking of anisotropy of the weak layer does not have much sense, since this layer is too thin to define a homogenized tensorial constitutive behavior. This is why we only investigate its response in a simple shear mode as it is usually done. Note also that the failure behavior of the weak layer strongly depends on the loading angle (Fig. 3), similar to what is observed in laboratory experiments (Reiweger et al., 2015).

**Comment M12).** *Figure 3 Perhaps call normal stress the "slope normal stress". The normal stress value, as well as shear, is a function of the coordinate system chosen.*
**Answer to comment M12).** Thanks for the suggestions. This will be changed throughout the whole manuscript.

**Comment M13).** *Caption You should state that compression is taken as positive.*

**Answer to comment M13).** This will be stated.

**Comment M14).** *Caption $\psi$ is labeled in the text as the slope angle. Should this be a critical slope angle? Should this be $\psi_t$ instead?*

**Answer to comment M14).** Thanks for pointing that out. $\psi$ is indeed the slope angle (or loading angle). $\psi_t$ is the angle for which we have a transition between the Mohr-Coulomb and Cap behavior (Reiweger et al., 2015). However it is written in the caption that $\tan \psi = \tau_p/\sigma_n$ which would indeed characterize a critical slope angle. This will be changed to $\tan \psi = \tau_g/\sigma_n$.

**Comment M15).** *Line 115 I'm not clear why you can state that these are satisfactory based on fig 3? While the initial slopes of increasing normal stress are similar, the actual values are quite dissimilar. Normal stresses are actually of opposite sign for the initial slope on the figure?*

**Answer to comment M15).** We do not claim any quantitative satisfactory reproduction here. We just say that the main features of real failure envelopes are modeled with our simplified weak layer, i.e. that we can have, tensile, shear, compression and mixed-mode failures in contrast to previous work which generally assumed only pure shear. This sentence will be reworded as follow: "the main features of real WL failure envelopes (Reiweger et al., 2015) are modeled with possible failures both in shear and compression (closed envelope)".

**Comment M16).** *Line 125 "specific fracture energy" is used only in the anticrack model. Correct? "the penetration resistance using the snow micropenetrometer (SMP) in the weak layer according to Reuter et al." This would not be possible if the layer was actually rigid, as is assumed for the anticrack model. Resistance would be infinite.*

**Answer to comment M16).** In our paper, we use the specific fracture energy only to compute the critical crack length from the anticrack model to compare to the result of our new formulation (Fig. 6). It was recently shown (Reuter et al., 2015) that integrating the penetration resistance over the weak layer thickness was leading to a good proxi of the specific fracture energy, although the physical assumptions are indeed not compatible. This will be clarified.

**Comment M17).** *Eq 1 $\psi_t$ is not defined.*

**Answer to comment M17).** $\psi_t$ is defined below eq. (4). It is the angle for which we have a transition between the Mohr-Coulomb and Cap behavior (Reiweger et al., 2015). We will try to define it earlier in the text.

**Comment M18).** *Line 140 So is K = 1 kPa ? From figure*

**Answer to comment M18).** Yes, this is correct, it will be added in the main text.

**Comment M19).** *Eq 5 Is this for the slab only? Or both slab and WL?*

**Answer to comment M19).** Scapozza's relationship between elastic modulus and density is used to describe the slab. This relationship has been derived for snow that is typically found in slab layers. The WL density is not involved in the model. This will be clarified by modifying the sentence line 145 as follows: "The Young's modulus of the slab was derived from density according to Scapozza (2004)".

**Comment M20).** *Line 148 Is this Poisson's ratio for the slab and the weak layer? Is this used for calculation of the elastic mismatch? Is the elastic mismatch used in your development or merely mentioned as an aside?*

**Answer to comment M20).** The Poisson's ratio $\nu$ is for the slab. This will be clarified as well. The Poisson's ratio is used for the calculation of the elastic mismatch, in detail $\Lambda = (E'DD_{wl}/G_{wl})^{1/2}$ with $E' = E/(1-\nu^2)$, as written line 171.

**Comment M21).** *Line 151 I presume that you mean to relate the tension in the slab to imply the slope parallel normal stress due to gravity. Slab bending may also cause a slope parallel tension component as well compression and shear in this same coordinate orientation*

**Answer to comment M21).** We agree, the slope parallel normal stress (tensile stress) in the slab is a combination of a pure tension term and a pure bending term ($\sigma_{xx} = \rho g a \sin\psi + 3\rho g \cos\psi a^2 / D$ according to beam theory). We would like to keep this terminology since it relates to the formulation of $\tau_{max}$ as the sum of a tension and bending term but we will change the sentence as follows: "...induces tension and bending in the slab...". We also agree that slab bending induces shear stresses in the slab which could in turn induce normal stress variations in the WL. Nevertheless the good agreement between our model (Eq. 9) and DEM results shows that these effects are of second order. A very detailed analysis of the stresses within the slab and the WL before the onset of crack propagation is currently undertaken in our group using the finite element method but is beyond the scope of the present study. To account for the reviewer's concern, we will add the following paragraph in the Discussion section: "Note that slab bending also induces shear stresses within the slab leading to possible normal stress variations in the WL. However, the good agreement between our model (Eq. 9) and DEM results (Fig. 4) suggests that these effects are of second order, and constitutes a validation of the assumption that the maximum shear stress at the crack tip has two main contributions related to slab tension and bending (Eq. 6)."

**Comment M22).** *Line 152 Is the crack tip calculated to be the thickness of the weak layer or at the bonds between the slab and WL? If it is at the interface, as appears to be implied at some points in the presentation, it would seem that this may differ from the max within the WL.*

**Answer to comment M22).** As stated before, due to the low density of the weak layer and its thin character, we assume the stresses in the weak layer to be homogeneous and due to the slab load as it is usually done. We hope that the changes made with regards to comment M8 will clarify this issue.

**Comment M23).** *Figure 4 Young's Modulus, E, is a function of density, $\rho$ in equation 5. Figure 4a has the critical crack varying with E with a fixed density of 300 kg m3. Similarly, figure 4b has a fixed E with crack length varying with density. Something seems wrong here? Is this somehow related to the calculation of E described in line 112?*

**Answer to comment M23).** Concerning this point, perhaps we were not clear enough about the use of Scapozza's relationship. This point also relates to comment M10. Scapozza's relation was introduced in the section called "Comparison with PST experimeents". For the DEM simulations, Scapozza's relation was not used. We made a systematic parametric study of the effect of each model property, with constant values of the other parameters in order to infer an analytical expression (this is stated lines 156-158 "...properties were varied independently in the simulations"). Then, once the analytical expression was found, we applied it to field data for which we did not measure directly the elastic modulus of the slab which was thus estimated using a relation based on Scapozza's laboratory experiments. To clarify this issue we will change the following sentence line 144: "The Young's modulus of the slab, which was not measured, was derived from density according to Scapozza (2004)". As mentionned above, this is not related to the statement in line 112 which explains how the grain properties are related to the macroscopic properties in the DEM model (Gaume et al., 2015).

**Comment M24).** *Line 163 Is this statement implying that you are considering only the influence of shear stress on the stress concentration? Relating to failure envelope? Again, is this assumed to be at the interface between the layers or within the WL itself?*

**Answer to comment M24).** We decided to present our model with shear stresses in order to relate our work to previous studies. However our model has a full mixed-mode shear-compression propagation criterion and could have been presented in compression as well. In a pure shear model (Eq. 7), the maximum stress is a function of the shear stress only. Our model has a compressive term $\sigma_n$ due to the addition of the bending term (Eq. 8) and because of the mixed-mode failure envelope allowing for compressive failure. It would be possible to propose a fully equivalent expression of Eq. (9) with a maximum compressive stress $\sigma_{max}$ but it would be harder to interpret it with regards to previous studies. The failure occurs inside the WL, not at the interface between the slab and the WL (but often in WL bonds near the interface).

**Comment M25).** *Equation 7 You are examining the case in which "WL has no thickness". But at this point $\Lambda$ (line 171) goes to zero so $a/\Lambda$ goes to infinity. The max shear in the WL then goes to infinity in equation 7? There is a singularity at the crack tip. This is also apparent in the bending component in equation 8. Even for any very thin layer of finite thickness this will be huge. This needs to be discussed and explained.*

**Answer to comment M25).** This is an important observation. We intended not to go into the details of the limiting case of zero thickness for the sake of simplicity but we will clarify this issue in the revised version. As mentioned above, Figure 4 is the result of a parametric analysis in which only one parameter was varied for each plot. However, for Figure 4d, when the WL thickness tends to zero, the weak layer shear modulus $G_{wl}$ becomes undefined since it is a volumetric quantity. Interfacial WLs are not characterized by the shear modulus $G_{wl}$ but by the shear stiffness $k_{wl} = G_{wl}/D_{wl}$. Hence the characteristic length becomes $\Lambda = (E'D/k_{wl})^{1/2}$ and is thus defined even for $D_{wl} = 0$. In Figure 4d, we did 3 simulations for different WL thicknesses and we kept the same shear modulus which explains the modeled trend, but in practice, having a critical crack length equal to zero and a fixed $G_{wl}$ would imply an infinite stiffness. We will add the following paragraph line 172 to explain this: "Note that in the limit case of a WL without thickness $D_{wl} = 0$, the WL shear modulus $G_{wl}$ becomes undefined since it is a volumetric quantity. Interfacial WLs are not characterized by the shear modulus $G_{wl}$ but by the shear stiffness $k_{wl} = G_{wl}/D_{wl}$ (Chiaia et al., 2008; Gaume et al., 2013). Hence, the characteristic length $\Lambda = (E'D/k_{wl})^{1/2}$ is still defined for $D_{wl} \rightarrow 0$."

**Comment M26).** *Line 177 suggest slope normal stress.*
**Answer to comment M26).** This will be changed as suggested.

**Comment M27).** *Line 180 If you assume a rigid weak layer, the elastic properties of the weak layer are irrelevant. It is also not clear why the rigid layer is assumed if the shear stress is independent of these properties.*
**Answer to comment M27).** We regret that this statement was confusing. We did not assume a rigid weak layer. The elastic character of the WL is one (together with the mixed-mode failure criterion) of the major improvements of our model compared to the anticrack model. Here we wanted to say that, in contrast to our model, if beam theory was used with a fixed boundary condition to represent the weak layer, the bending term would scale with $D$ instead of $\Lambda$ similar to the anticrack model since it assumes a rigid weak layer. This sentence will be reworded accordingly to avoid any confusion: "If, in contrast to our model, beam theory (Timoshenko and Goodier, 1970) was used with a fixed boundary condition to represent the WL to compute the bending term $\tau_{max}^b$, it would scale with $\sigma_n(a/D)^2$, similar to the anticrack model since it assumes a rigid weak layer."

**Comment M28).** *Line 180 I think you need to more explicitly describe the beam boundary conditions assumed. according to beam theory is rather vague. I am also not seeing where this reference (Timoshenko and Goodier, 1970), is dealing with cracks, except for a brief mention relative to Griffith fracture. Can you be more specific?*
**Answer to comment M28).** The boundary conditions will now be described (see previous point). In addition, we will add to the paper (see above), that beam theory is used only to compute the bending term in the case of a beam bending over a fixed substratum and compare it to our formulation based on DEM. This is not a crucial point but allows to emphasize the importance of considering the elastic mismatch.

**Comment M29).** *Line 182 suggest replace allows with provides a means*
**Answer to comment M29).** This will be changed as suggested.

**Comment M30).** *Line 182 Is it using Dundur's elastic mismatch parameters in the numerical solution? Does this imply that you are calculating the crack to be at the interface between layers? How does scaling with $\lambda$ instead of D come into play?*

**Answer to comment M30).** Our characteristic length $\Lambda$ is not the same as the Dundurs' elastic mismatch parameters. You can find in Chiaia et al. (2008) and Gaume et al. (2013) the full derivation of $\Lambda$. However, it is likely that it is possible to relate $\Lambda$ to the Dundurs' parameters and the ratio $D/D_{wl}$ since $\Lambda$ can also be expressed as

$$\Lambda = D \frac{\sqrt{E'/G_{wl}}}{\sqrt{D/D_{wl}}}.$$

Hence, the crack tip is not supposed to be at the slab - WL interface, but within the WL.

Concerning the second part of the comment, the scaling with $\Lambda$ instead of $D$ is extremely important. Figure 5 highlights this crucial part of our paper. Any scaling with $D$ could not explain the DEM results since it would not account for the elastic mismatch between the slab and the weak layer. The clarifications that will be made to the paper (comment M27) will also make this point clearer.

**Comment M31).** *Line 191 I presume that the residual stress, r, is not used as the result of this condition. And this paper is not concerned with the sliding following failure of the week layer.*

**Answer to comment M31).** That is perfectly correct. If this condition was not fulfilled, one would need to correct $\tau_{max}$ by removing a term related to this frictional stress (see comment M9).

**Comment M32).** *Line 211 This is an important result, since in the avalanche community the concept of slope independence has been quite widely accepted of late.*

**Answer to comment M32).** Thanks for this remark, we agree that this is important but may be seen as controversial according to Reviewer #2. We provide a detailed answer on this issue with explanations of the discrepancies between our model and trends observed in field experiments in the replies to Reviewer #2.

**Comment M33).** *Line 215 Here again I think that it should be clearly stated that you are referring to the stress due to the gravity along the slope and independently, the stress state as the result of bending.*

**Answer to comment M33).** See comment 21. The sentence line 215 will be changed to "...the interplay between tension and bending in the slab.".

**Comment M34).** *Line 215 Parts of this development seem to imply that the failure would be at the bonds between the slab and the weak layer. I think that what is being calculated should be clearly stated. As an aside, in the physical situation for depth hoar the failure may well be at the lower interface or at the upper and lower. The crystals themselves often appear to remain intact.*

**Answer to comment M34).** Please refer to answers on comments M8, M22 and M24. In addition, if the failure would be at the interface only, the WL thickness would play no role (see also comment M25) similar to the anticrack model, which is not the case here.

**Comment M35).** *Line 221 suggest changing accounts for slab bending only to accounts for stresses due to slab bending only*

**Answer to comment M35).** This will be done as suggested.

**Comment M36).** *Line 223 Should clearly state that you are referring to the shear stress at the crack tip induced by the slope parallel loading of the slab. This results in the shear stress at the interface between the slab and WL.*

**Answer to comment M36).** This sentence will be changed as follows: "For steep slopes ($\psi > 30°$), where the shear stress at the crack tip due to slab bending becomes negligible compared to that due to slab tension,...".

**Comment M37).** *Line 224 ac t goes toward zero as WL thickness goes to zero. Following discussion above.*

**Answer to comment M37).** Please refer to answer on comment M25.

**Comment M38).** *Figure 6 caption suggest changing represents to represent*
*Line 227 suggest changing decreases to decrease.*

**Answer to comment M38).**    Thanks, this will be changed.

**Comment M39).**    *Line 252  I think that you should state that you are using a mechanics of materials or perhaps an elasticity approach, if that is appropriate.*
**Answer to comment M39).**    The sentence line "252 will be modified as follows."...from which analytical expressions can be inferred using a strength-of-material approach."

**Comment M40).**    *Line 254 Suggest changing allows to reconcile shear and collapse to reconciles the shear and collapse*
**Answer to comment M40).**    The will modified as suggested.

**Comment M41).**    *Line 321  suggest changing that skier triggered avalanches are more likely on steep rather than on flat slopes to that avalanche initiation is more likely on steep rather than on flat slopes*
**Answer to comment M41).**    As we generally use the term initiation in the sequence of processes preceding avalanche release (failure initiation and then crack propagation), we will modify this sentence as follows: "...triggering an initial failure leading to slab avalanche release is more likely on steep rather than on flat slopes".

**Reply to Referee #2**

We thank Referee #2 for his constructive comments that helped us to significantly improve the quality of our paper, in particular, by providing a more detailed discussion about the effect of slope angle and comparison with field experiments.

We provide below detailed answers to the various issues raised by the reviewer and we will revise our paper in order to account for his remarks.

**Main comments**

**Comment M1).**

*I am not a DEM expert, but the model set up and results appear sound to me. The analytical work is also sound. My main criticism is that the analytical model does not explain previous fieldwork with PSTs [Gauthier and Jamieson, 2008; McClung,2009; Bair et al., 2012] showing that snowpacks with identical slab thickness (slope normal) and densities showed the same or increasing critical cut lengths as slope angle increased. Rather than addressing this discrepancy directly, the authors dismiss previous fieldwork with PSTs by stating that the snowpack properties could not have remained the same throughout the range of slope angles tested, thereby calling into question the accuracy of the reported values in each of these studies. As the author of one of these previous field studies, I find this particularly irksome given the trouble that I went to in ensuring uniform conditions throughout the slope angles tested. Further, the authors do not present any field evidence to support their modeled slope angle dependence, i.e. PSTs showing decreasing critical length with increasing slope angle given constant slab thickness and density. Finding constant slab thickness and density over a range of slope angles in the field is not that difficult, which explains why multiple studies have been able to do it.*

**Answer to comment M1).**

We agree that we did not discuss in detail the discrepancy between the model prediction and the results of previous field work. We will do so in the revised manuscript.

Whereas we do not question that the authors of the mentioned field work tried to perform their experiments as good as possible under uniform conditions, we are not aware that the WL mechanical properties (shear strength or specific fracture energy) as well as slab properties were quantified in detail to actually confirm uniform conditions in space and time. In fact, the slab thickness (slope normal) decreases with slope angle in Bair et al. (2012) in contrast to what is stated by the reviewer (cf section 3.2 of their paper). Furthermore, the weak layer properties (which were not measured in these studies) might be significantly different with respect to slope angle due to different settlement/creep processes between a flat, undisturbed snowpack and a sloping one and due to the inherent spatial variability of the snowpack (Schweizer et al., 2008).

In addition, in the study of Gauthier and Jamieson (2008), according the PhD thesis of Dave Gauthier (page 104, Gauthier, 2007) the measurements made at 0 degrees were performed one day before the measurements that were made on slopes which would lead to important differences in WL strength due to settlement and sintering processes since the WL consisted of precipitation particles and which would in turn strongly influence the critical crack length (Brown et al., 2001; Szabo and Schneebeli, 2007; Podolskiy et al., 2014). Just as an example, in Bair et al. (2012), although the slab remained the same between the 19 and 20 March 2011, the weak layer hand hardness index changed from 0.8 to 1. In fact, in both Gauthier and Jamieson (2008); Bair et al. (2012), the WLs are composed of precipitation particles, not typical persistent WLs.

Before going into the explanation of the discrepancies between our results and the outcomes of the field studies of Gauthier and Jamieson (2008); Bair et al. (2012), we would like to point out that previous comparisons of the anticrack model with field experiments include some ambiguities due to the type of presentation (log-scale, marker size). For example, let's consider Fig. 3 in Bair et al. (2012) more closely, since more data are available at different slope angles and because the data were gathered the same day (19 March 2011). We digitized their data and plotted them on Fig. 1 of the document. First of all, it seems that there is a good overall

agreement with the anticrack model. We want to highlight the fact that, in their study, the WL specific fracture energy and the elastic modulus of the slab were derived (and not measured) from a fit of the anticrack model to the data (this method was developed by the second author of our paper: van Herwijnen and Heierli (2010); van Herwijnen et al. (2016)). Hence, in the end, it is normal that a relatively (log-scale, large markers) good agreement is observed since the only parameter characterizing the WL and the slab elastic modulus were fitted to the data. If we plot, with the same axis and log scale as in Bair et al. (2012), our model (with the same input properties as in Bair et al. (2012) and a cohesion of 500 Pa estimated from hand hardness and grain type according to Geldsetzer and Jamieson (2001); Jamieson and Johnston (2001)) next to their data (Fig. 1a), we would not be able to say which of the models is the best. In fact, even a pure shear model (Eq. 7 of our paper) with the same cohesion (500 Pa) would lead to a reasonable agreement. This is not really the way we wanted to validate our model. In contrast, in our paper, when we compare our model to the data (Fig. 6) none of the mechanical properties are fitted, and still, we find a better agreement than the anticrack model so we do not agree that our model does not explain field data, on the contrary.

In addition, in Fig 1a, our model (Eq. 9) shows a gentler decrease of the critical crack length with increasing slope angle compared to Fig. 4 of the paper. In fact, in this example, the critical crack length becomes almost constant for $\psi > 25$ degrees with our model. There are two reasons for this behavior:

- The slab density in Bair et al. (2012) is extremely low (84 kg/m$^3$, storm snow) leading to a very low elastic modulus. Fig. 2 below shows that the slab elastic modulus has a strong effect on the decrease of the critical crack length with slope angle, the lower the modulus, the lower the decrease.

- In the data of Bair et al. (2012), the slab depth $H$ is constant as usually observed (gravity-driven snow-falls). Hence, the slab thickness $D$ decreases with increasing slope angle according to $D = H \cos \psi$. Since a lower slab thickness leads to a higher critical crack length, this explains the gentler decrease. Note that this variation in slab thickness with slope angle was correctly accounted for by Bair et al. (2012) in the comparison with the anticrack model but was seemingly disregarded in Heierli et al. (2008) who assumes a constant (and extremely low) slab thickness $D = 0.11$ m.

[Figure]

Figure 1: (a) Critical crack length vs slope angle: comparison between the data of Bair et al. (2012) (black circles) and our new model (Eq. 9, red line), the anticrack model (purple dashed-line) and a pure shear model (Eq. 7, green dotted line) for a constant slab depth $H = 0.35$ m ($D = H \cos \psi$) and same input properties as in Bair et al. (2012) with a semi-log scale. The cohesion $c = 500$ Pa was estimated for the hand hardness provided in Bair et al. (2012) using Geldsetzer and Jamieson (2001); Jamieson and Johnston (2001). Inset: linear scale. (b) Effect of geometry on the slope angle dependency. SNF: Slope normal faces. VF: Vertical faces.

In this example, we managed to recover the trend in the field only by accounting for variations in slab thickness. We would probably get an even better agreement if we knew the variations in WL shear strength. This is exactly as we wrote in our paper "Hence we argue that the dependence of the critical crack length with slope angle obtained with a model with constant values of the other parameters should not be compared to

trend observed in the experiments which is the result of a combination of many varying properties". Similarly, if we account for the vertical character of the PSTs in Gauthier and Jamieson (2008) by adding $D/2 \tan \psi$ to the critical length (Heierli et al., 2008), we would actually obtain a slightly increasing trend which nicely reproduces their data (Fig. 1b). However, as stated in the paper, we prefer to show the true trend with slope angle and make a meaningful point by point comparison with field data as it is done in our paper. Nevertheless, a new paragraph (provided in the appendix) and three new graphs (Fig. 1 and 2 of this document) will be added to our paper to answer the reviewer's concern.

[Figure]

Figure 2: Effect of the slab elastic modulus on the slope angle dependency of the critical crack length for $\rho = 200$ kg/m³, $D = 0.2$ m, $D_{wl} = 4$ cm. Inset: Effect of the slab depth on the slope angle dependency of the critical crack length for $\rho = 200$ kg/m³, $E = 2$ MPa, $D_{wl} = 4$ cm.

**Comment M2).** *Note that the same behavior has been shown in Extended Column Tests [Heierli et al., 2011; Bair et al., 2012] and Compression Tests [Birkeland et al., 2014], all with the same slab thickness and density. I realize those tests were not modelled here, but the fact that the three main stability tests all show the same trend cannot be ignored.*

**Answer to comment M2).** We do not ignore these results. First, the ECT is made with vertical faces such as the PSTs measurements from Gauthier and Jamieson (2008); Heierli et al. (2008). Our model and data have slope-normal faces. In line with the vertical to slope-normal correction proposed by Heierli et al. (2008), if the ECT score remains constant with increasing slope angle it means that you would get a decreasing trend with slope normal faces. This is again in line with our model results. Second, when the effect of slope angle on the ECT score is studied, the slab depth is generally constant, we recall that in our model, the slab thickness (slope normal) is constant. If one accounts for both vertical faces and the decrease of slab depth with increasing slope angle in our model, this would lead to a slightly increasing trend with increasing slope angle as shown in Fig. 1b (blue dashed-dotted line). Finally it is important to note that the loading conditions (vertical taps) in the CT/ECT lead to non-uniform loading conditions with respect to slope angle.

**Comment M3).** *Perhaps the authors need to re-read these studies more carefully. For instance, they state that Heierli et al. [2008] only used three PSTs to validate the anticrack model. In fact, it was 44 PSTs over three different slope angles [Gauthier and Jamieson, 2008] used, with the means taken for each of the three slope angles.*

**Answer to comment M3).** In fact, we read this paper in the greatest detail, even the supplementary material, which allowed us to note that Heierli et al. (2008) did not account for the vertical character of the PST faces in the data comparison. We can find no evidence that Heierli et al. (2008) used the 44 PSTs of Gauthier and Jamieson (2008). Looking at the data of Gauthier and Jamieson (2008) (Table on Fig. 3) and that of Heierli et al. (2008) (Table in Fig. 4), one realizes that the data are partly different: the average slab thickness at

zero degree (i.e. $D = H \cos 0 = H$) is 0.14 m in Gauthier and Jamieson (2008) and 0.11 m in Heierli et al. (2008); the median critical crack length for 0 degree is 0.24 m in Gauthier and Jamieson (2008) and 0.13 m in Heierli et al. (2008) (it is not a matter of correction since $\tan 0 = 0$); the median critical crack length for 38 degrees is 0.13 m in Gauthier and Jamieson (2008) and 0.22 m in Heierli et al. (2008). So it is actually not very clear which subset of the data was used. In addition, Heierli et al. (2008) assume a constant slab thickness $D$ although the depth $H$ should be constant, as correctly assumed in the paper of Bair et al. (2012). Finally, (Gauthier, 2007, p. 104) performed the 23 measurements made at zero degree one day before the 21 tests made on slopes (17 at 28-30 degrees and 4 at 38 degrees) in contrast to what is shown in Gauthier and Jamieson (2008) and Heierli et al. (2008). This indicates that the trend with slope angle may be flawed since sintering and settlement effects can strongly affect snowpack properties within one day, especially with the layer of precipitation particles which was tested. Again, we do not question the comprehensive and very important field work of Gauthier and Jamieson (2008), only the use of their data in Heierli et al. (2008).

However, the above mentioned discrepancies are not really essential, but we simply pointed them out in response to the reviewer's criticism. We will not go into these details in the revised manuscript as we prefer to focus on our new model rather than on the ambiguities in previous work, but we hope that the new paragraph on slope angle dependency (see Appendix) will provide the clarification needed.

Table 1
Summary of experiment days, including selected results snowpack properties

| Date | $n$ | Cut dir. | Type | $\psi$ (°) | Cut (m) | Slab | | Weak layer | |
|---|---|---|---|---|---|---|---|---|---|
| | | | | | | $H$ (m) | $\rho$ (kg m$^{-3}$) | Form | $E$ (mm) |
| 24-JAN-06 | 23 | ~ | *CrL* | 0 | | | | | |
| | 17 | *UP* | *CrL* | 29–30 | | | | | |
| | 4 | *UP* | *CrL* | 38 | 0.13 | 0.14 | 134 | 2a | 2–4 |
| **24-Jan-06** | **23** | **~** | **CrL** | **0** | **0.24** | **0.14** | **134** | **2a** | **2-4** |
| | **17** | **UP** | **CrL** | **29-30** | | | | | |
| | **4** | **UP** | **CrL** | **38** | | | | | |

Figure 3: Table taken from Gauthier and Jamieson (2008) (top) and a preprint of Gauthier and Jamieson (2008) (bottom) showing some snowpack properties on 24 January 2006, when the slope angle dependence was studied. Note that $H$ corresponds to slab depth (measured vertically).

| | $\rho$ (kgm$^{-3}$) | $E$ (MPa) | $h$ (cm) | $\theta$ (deg) | $r_c$ (cm) | $w_f$ (J/m$^2$) | Cut direction |
|---|---|---|---|---|---|---|---|
| Gauthier *et al.* (9) | 134 | 1.5 ± 0.8 | 11 | 0 | 13 | 0.03 | Up |
| | | | | 30 | 19 | | Up |
| | | | | 38 | 22 | | Up |

Figure 4: Table taken from Heierli et al. (2008) showing the data used for their model validation. Note that $h$ represents the slab thickness (slope normal)

**Comment M4).** *Likewise, one of the major criticisms of the anticrack model has been that it assumes a linear elastic slab. This model is supposed to be an improvement in accuracy, but the authors make the same linear elastic assumption without discussing pitfalls. This may explain some of the scatter in Figure 6 for both models.*

**Answer to comment M4).** Our model is an improvement of the anticrack model in many ways: (i) slope-dependent failure criterion of the weak layer (as observed and measured in laboratory experiments, Reiweger et al. (2015); Chandel et al. (2015)), (ii) elasticity of the weak layer treated as a real material with finite thickness, (iii) accurate modeling of the interplay between slab bending and tension based on DEM simulations. Our DEM model allows for failure of the bonds in the slab leading to an elastic-brittle behavior. We recently showed (Gaume et al., 2015) that the propagation distance (distance between the lower edge and slab fracture) was almost always higher than the critical crack length except for combinations of very low slab densities and

thicknesses. This behavior is also observed in field experiments. Accordingly, the linear elastic assumption for the slab makes sense. This assumption was also confirmed by recent field studies (van Herwijnen and Heierli, 2010; van Herwijnen et al., 2016) showing that the slab displacement measured using particle tracking can be described by beam theory with a linear elastic assumption. In addition, we believe that a slab thickness of 0.11 m (Heierli et al., 2008) is extremely low and thus rather the exception than the rule for which size effects might be very important. Concerning the scattering in Figure 6 it is very likely due to the uniform character of the slab which is assumed in our model as shown by Reuter et al. (2015) (Fig. 8 of their paper) who managed to significantly reduce the scattering by accounting for the layers in the slab. This is stated lines 265-272 of our paper. To account for the reviewer's concern about the linear elastic assumption, we will add the following sentences to our revised paper in section 4. Discussion - Relevance and limitation:

"In a recent study, Gaume et al. (2015) showed that the DEM model can reproduce the dynamic phase of crack propagation as well as fracture arrest in the slab which was treated as an elastic-brittle material. In particular, the crack propagation speed and distances obtained by PTV analysis of the PST were well reproduced. It was also shown that the propagation distance (distance between the lower edge and slab fracture) was almost always higher than the critical crack length except for combinations of very low slab densities and thicknesses. This behavior is also observed in field experiments. Accordingly, treating the slab as a linear elastic material before the onset of crack propagation is justified. This assumption was also confirmed by recent field studies (van Herwijnen and Heierli, 2010; van Herwijnen et al., 2016) showing that the slab displacement as obtained particle tracking can be described by beam theory with a linear elastic assumption. Hence, with the present study, we show that our model is able to address the whole crack propagation process.".

**Comment M5).**    *Further, the evaluation of the anticrack model is flawed since a constant (0.1 J m-2) specific fracture energy is used. Field data show this value varies by more than an order of magnitude [Schweizer et al., 2011]. A constant specific fracture energy in the anticrack model is akin to setting a constant value for the weak layer shear strength in the authors model.*

**Answer to comment M5).**    We do not agree on this point. The anticrack model inherently considers that the failure of the WL is slope independent which is why the weak layer fracture energy was not modified with respect to slope angle exactly in the way it was proposed in Heierli et al. (2008). This is the very nature of the anticrack model. A constant specific fracture energy is akin to a unique failure envelope of the weak layer in our model which is what was set (constant cohesion, tensile and compressive strength). The slope-dependent failure criterion is an improvement of our model. One could of course try to improve the anticrack model by assuming a slope-dependent failure behavior of the WL but this is not the point here. However, we did not assume a constant specific fracture energy of the WL when we compared field data to the anticrack model. The WL specific fracture energy was evaluated from the SMP resistance (Reuter et al., 2015) and ranges from 0.07 to 2.9 J/m$^2$ and effectively span two orders of magnitude.

**Comment M6).**    *The manuscript requires extensive editing to correct grammatical errors. I suggest an English language service. For instance, there were too many errors of tense for me to correct. I gave up. Likewise, symbols are used in graphs and equations without being defined until later in the text.*

**Answer to comment M6).**    Thanks for these corrections. We will improve the language and correct remaining grammatical errors.

**Comment M7).**    *I suggest deleting the Application to simulated snow stratigraphy section, as it is not convincing with only one measured and modelled profile for comparison. Other minor corrections are included as an annotated PDF*

**Answer to comment M7).**    The SNOWPACK implementation highlights that our new formulation is directly applicable in physics based snowpack models and produces useful results. This shows the relevance and applicability of our work. Of course, the example we provide should by no means be seen as a validation, but rather as an illustration of the potential of our new model and future prospect. After thorough validation, the implementation of our new model into SNOWPACK will very likely lead to the improvement of avalanche forecasting by accounting for crack propagation and not only classical stability indices in the evaluation of the

avalanche danger, as suggested by (Reuter et al., 2015). This application also stresses that our model is implemented in SNOWPACK and available to everyone for download for research or for practical purposes. As a consequence, we prefer to keep this section.

To account for the reviewer's concern, we will change the conclusion sentence as follows:"While validation is still required, this opens promising perspectives to improve instability evaluation by combining traditional stability indices with a new metric to evaluate crack propagation propensity."

**Specific comments**

**Comment m1).** *line 5: better explain instead of allow for better explaining.*
**Answer to comment m1).** This change will be done.

**Comment m2).** *Line 14: change to opens a promising*
**Answer to comment m2).** This change will be done.

**Comment m3).** *Line 16: rank instead of range*
**Answer to comment m3).** This change will be done.

**Comment m4).** *Line 18: remove ubiquitous*
**Answer to comment m4).** This change will be done.

**Comment m5).** *Line 18: remove ubiquitous*
**Answer to comment m5).** This change will be done.

**Comment m6).** *Line 32: the critical crack length instead of critical crack length.*
**Answer to comment m6).** The Editor suggested to remove the "the" article as much as possible. However, we will reconsider this.

**Comment m7).** *Line 40: more highly resolved spatially? Temporally? Geographically? All of the above?*
**Answer to comment m7).** Spatially and temporally. This will be specified.

**Comment m8).** *Line 64: That's the definition of fracture toughness.*
**Answer to comment m8).** We agree, however, the fracture toughness is related to the specific fracture energy through the elastic modulus. We feel that saying that the specific fracture energy represents the resistance to crack propagation is more illustrative than saying it is the energy dissipated during fracture, which would be actually wrong for the anticrack model since no energy is dissipated as the weak layer is purely rigid.

**Comment m9).** *Line 106: I assume modeling spherical elements at the size of snow grains themselves (e.g. r=0.0001) would be too costly computationally? What's the tradeoff?*
**Answer to comment m9).** This would have been indeed more costly computationally but this is not really the point. We could have grains with r=0.00001m but with the same macroscopic failure envelope whose shape depends only on the WL structure. Hence, the tradeoff is to have grain radii smaller than the WL thickness to be able to obtain the desired WL structure. This point was addressed in detail by Gaume et al. (2015).

**Comment m10).** *This caption needs refer to each of the elements (a-e) individually. Many of the symbols, e.g. $l_0$ and Lambda, have not been defined in the text yet.*
**Answer to comment m10).** This will be done as suggested.

**Comment m11).** *Can you explain this further? It looks like upside down surface hoar. How was the triangular shape chosen?*

**Answer to comment m11).**   The shape of the WL was chosen to mimic the porous and unstable nature of weak layers consisting of depth hoar or surface hoar. This allows to obtain a realistic closed failure envelope (including tensile, shear, compressive and mixed-mode failure). The angle of the crystal is very important and characterizes the slope of the increasing and decreasing parts of the failure envelope as shown in Gaume et al. (2014). However, the crystals being close to each other, this structure is mechanically equivalent to the same structure with upside down structures. Finally, note that our expression of critical crack length (Eq. 9) is independent of the WL structure since one can input any failure criterion in $\tau_p$.

**Comment m12).**   *Can you comment on the differences in modeled vs observed shear strength? Where does the mismatch arise from?*
**Answer to comment m12).**   Our objective was not to obtain the exact same failure envelope as Reiweger et al. (2015) but to obtain similar important features such as a mixed-mode shear compression failure, which is important for failure initiation but also for crack propagation. Note that a WL made from ballistic deposition would lead to a quantitatively similar failure envelope as in Reiweger et al. (2015) as shown by Gaume et al. (2014). Please refer to Gaume et al. (2014) for more details about the failure envelope.

**Comment m13).**   *Line 139 Lambda is not defined*
**Answer to comment m13).**   Right, $\Lambda$ will be defined here.

**Comment m14).**   *line 145 Please put units here and elsewhere You need to discuss why the linear assumption was made, given that snow, like most materials, is not purely elastic under high strain rates.*
**Answer to comment m14).**   We will add the units here since it is an empirical relationship. For the other relationships, being analytical, there is no need to precise the unit. As discussed in the major comments above, a new paragraph will be added to the Discussion section in which the linear elastic assumption will be justified.

**Comment m15).**   *line 147 What Eq. is this used in?*
**Answer to comment m15).**   $G_{wl}$ and $\nu$ are used to compute $\Lambda$. We agree it was not clear since $\Lambda$ was only defined in section 3. $\Lambda$ will be defined earlier to clarify this issue.

**Comment m16).**   *Fig 4 This figure is shown to earlier. Eq 9 isn't shown until near the end of the next page, after Fig 5. Each plot in this figure needs to be labeled.*
**Answer to comment m16).**   We agree but there is more information in this figure than the results of Eq. 9. In particular, the grey markers are the DEM simulations results which are presented at the beginning of Section 3. It is quite usual to describe the results in several steps: 1. numerical model results 2. analytical model. The plots are already labeled from a to e.

**Comment m17).**   *Line 172 E' is given in Eq. 4 but not defined until here.*
**Answer to comment m17).**   We agree and this will be modified (see m13 and m15)

**Comment m18).**   *Fig 5 varied how? No ranges are given.*
**Answer to comment m18).**   The parameters were varied according to the ranges shown in Fig. 4. The points correspond to the same simulations shown in Figs. 4a 4b 4c and 4d (zero slope angle). This will be clarified.

**Comment m19).**   *line 191 remove "(not shown)"*
**Answer to comment m19).**   This will be done.

**Comment m20).**   *line 200 remove "As for the... (not shown)"*
**Answer to comment m20).**   We will remove "not shown" but we prefer to keep the first part. Indeed, we said above that the critical crack length was always lower than $l_0$ in the simulations. Here we say the same thing but for the experiments which is different and which strengthens this conclusion.

**Comment m21).** *Where does this come from? Using a constant value for wf almost ensures that the anticrack model will not perform as well.*

**Answer to comment m21).** Please refer to the major comment M5. We used a constant WL specific fracture energy only for the comparison to our DEM simulations for which we have one single failure envelope (constant cohesion, tensile and compressive strength). Having a specific fracture energy independent of the slope angle is in the inherent nature of the anticrack model. Furthermore, we are not aware of any studies on the effect of slope angle on the weak layer specific fracture energy; it is not our objective here to improve the anticrack model. As explained in Section 2 "Comparison with PST experiments", the WL specific fracture energy was estimated from SMP measurements and ranges from 0.07 to 2.9 $J/m^2$.

**Comment m22).** *Read Heierli et al. (2008) and its citations in Table 1 more carefully. The data used from Gauthier et al (2008) comprised means from 44 PSTs @ 3 different slope angles.*

**Answer to comment m22).** Please refer to comment M3.

**Comment m23).** *This argument dismisses careful field work done with the PST, all of which shows that $a_c$ is constant or increases with slope angle in PSTs (e.g. Gauthier and Jamieson, 2008; McClung, 2009; Bair et al. 2012) even when slab thickness (slope normal) and density were carefully measured and held constant. From your list, the only material property not measured in these tests was weak layer strength, which you claim decreases with slope angle (Reiweger and Schweizer, 2010; 2013). Thus, $r_c$ should have decreased with slope angle yet the opposite occurred in these careful field tests. I suggest you hypothesize on other reasons besides D/E, rho, and WL strength for the discrepancy.*

**Answer to comment m23).** Please refer to comment M1. Furthermore, we would like to point out that there is a big difference between the effect of the slope (or loading) angle on the same snowpack configuation (Reiweger and Schweizer, 2010; 2013; Reiweger, 2015) which we studied here and the effect of slope angle in reality. The settlement/creep of snow in steep slopes will induce different sintering effects than on a flat snowpack. So if you take different samples of the same snowpack and load them with different loading angles, you would find a decreasing trend of the strength or critical crack length with slope angle. However, if you take samples at different slope angles, even with the same slab properties, you might not have a decresing trend because the weak layer went through different settlement/sintering processes. The intrinsic mechanical behavior of a material needs to be distinguished from the physical process acting at the slope scale. This is why we prefer to make a point-to-point comparison in which the properties measured at each snowpit are used directly as input of our model. Finally we want to recall that the slab thickness is not constant in Bair et al. (2012) but decreasing with increasing slope angle (constant slab depth, cf section 3.2 of Bair et al. (2012)) contrary to what is stated by the reviewer.

**Comment m24).** *line 279 remove might*

**Answer to comment m24).** This will be done

**Comment m25).** *This section is unconvincing, given only one measured/modeled profile. I suggest it be deleted.*

**Answer to comment m25).** Please see comment M7.

**A    Appendix – New paragraph to be included in the Discussion section: Slope angle dependency**

We showed that the critical crack length $a_c$ decreases with increasing slope angle $\psi$ for a PST with slope-normal faces, a constant slab thickness $D$ and constant values of the other mechanical properties. Fig. A.1 shows that the rate of decrease of $a_c$ with $\psi$ is strongly influenced by the elastic modulus $E$ and thickness $D$ of the slab. Low values of $E$ and/or $D$ lead to a gentler decrease of $a_c$ with $\psi$.

[Figure]

Figure A.1: Effect of the slab elastic modulus on the slope angle dependency of the critical crack length (Eq. 9) for $\rho = 200$ kg/m$^3$, $D = 0.2$ m, $D_{wl} = 4$ cm. Inset: Effect of the slab depth on the slope angle dependency of the critical crack length for $\rho = 200$ kg/m$^3$, $E = 2$ MPa, $D_{wl} = 4$ cm.

However, in the field, slab thickness $D$ (slope normal) is not constant with respect to slope angle but slab depth $H$ (vertical) generally is. Accordingly, the slab thickness decreases with increasing slope angle according to $D = H \cos \psi$. Since a lower slab thickness leads to a higher critical crack length (Fig. 4c) this leads to a reduction of the decrease of $a_c$ with $\psi$. As an illustration, we compare on Fig. A.2 our model (Eq. 9) to the data of Bair et al. (2012) for which $H$ is constant and the slab density and elastic modulus are very low (storm snow, $\rho = 84$ kg/m$^3$, $E = 0.22$ MPa). The low elastic modulus and the decrease of slab thickness with $\psi$ thus lead (Eq. 9) to a very gentle decrease of $a_c$ with $\psi$ which reproduces well the data. The anticrack model was also plotted on Fig. A.2a and shows very comparable results. Yet, the values of the WL specific fracture energy $w_f$ and slab elastic modulus $E$ in Bair et al. (2012) were estimated by a fit of the anticrack model to the data using the method developed by van Herwijnen et al. (2010, 2016) which in turn, obviously leads to a good but meaningless agreement. Note also that even a pure shear model (Eq. 7) with the same input properties as for our model (Eq. 9) would lead to a reasonable agreement for steep slopes ($\psi > 30°$). In the studies of Heierli et al. (2008); Bair et al. (2012), the significant difference obtained between the anticrack model and the pure shear model (McClung, 1979; Gaume et al., 2013) is irrelevant because the same specific fracture energy was taken as input of both models although the underlying physical assumption are strictly incompatible. Indeed, the pure shear model considers a quasi-brittle behavior for the weak layer and the anticrack model considers a purely rigid one. In fact, for $\psi > 30°$ and short critical crack lengths which are typically encountered in field experiments, Gaume et al. (2014) recently showed from the energy balance equations that both approaches lead to very comparable results, which is confirmed by our new findings.

Finally, we want to point out the significant influence of geometrical effects on the slope angle dependency of the critical crack length. Figure A.2b shows the critical crack length vs slope angle for three different PST configurations: (i) constant slab thickness $D$ and slope normal faces (SNF); constant slab depth $H$ and and slope normal faces (SNF); (iii) constant slab depth and vertical faces (VF). The vertical character can be accounted for by adding $D/2 \tan \psi$ to the critical crack length as proposed by Heierli et al. (2008) (in the supplement). We clearly observe that the decrease of $a_c$ with $\psi$ is gentler with a constant slab depth $H$ than with a constant slab thickness $D$ as shown before. In addition, we observe an increase of the critical crack length with increasing

[Figure]

Figure A.2: (a) Critical crack length vs slope angle: comparison between the data of Bair et al. (2012) (black circles) and our new model (Eq. 9, red line), the anticrack model (purple dashed-line) and a pure shear model (Eq. 7, green dotted line) for a constant slab depth $H = 0.35$ m ($D = H \cos \psi$) and same input properties as in Bair et al. (2012) with a semi-log scale. The cohesion $c = 500$ Pa was estimated for the hand hardness provided in Bair et al. (2012) using Geldsetzer and Jamieson (2001); Jamieson and Johnston (2001). Inset: linear scale. (b) Effect of geometry on the slope angle dependency. SNF: Slope normal faces. VF: Vertical faces. const.: constant.

slope angle if the PST is made with vertical faces and if the slab depth is constant. This is in line with the PST experiments of Gauthier and Jamieson (2008) performed with vertical faces and a constant slab depth $H$. Note that Heierli et al. (2008) did not account neither for the vertical character of the faces nor the decrease of slab thickness with slope angle in their comparison to the data of Gauthier and Jamieson (2008). The increasing trend observed for our model with a constant slab depth $H$ and vertical faces might explain why the ECT (Extended Column Test) score is often increasing with increasing slope angle (Heierli et al., 2011; Bair et al., 2012).

**References**

Bair, E. H., R. Simenhois, K. Birkeland, and J. Dozier, 2012: A field study on failure of storm snow slab avalanches. *Cold Reg. Sci. Technol.*, **79**, 20–28.

Brown, R., P. Satyawali, M. Lehning, and P. Bartelt, 2001: Modeling the changes in microstructure of snow during metamorphism. *Cold regions science and technology*, **33**(2), 91–101.

Chandel, C., P. K. Srivastava, and P. Mahajan, 2015: Determination of failure envelope for faceted snow through numerical simulations. *Cold Regions Science and Technology*, **116**, 56–64.

Chiaia, B., P. Cornetti, and B. Frigo, 2008: Triggering of dry snow slab avalanches: stress versus fracture mechanical approach. *Cold Reg. Sci. Technol.*, **53**, 170–178.

Gaume, J., G. Chambon, N. Eckert, and M. Naaim, 2013: Influence of weak-layer heterogeneity on snow slab avalanche release: Application to the evaluation of avalanche release depths. *J. Glaciol.*, **59(215)**, 423–437.

Gaume, J., G. Chambon, I. Reiweger, A. van Herwijnen, and J. Schweizer, 2014: On the failure criterion of weak-snow layers using the discrete element method. *P. Haegeli (Editor), 2014 International Snow Science Workshop, Banff, Alberta.*

Gaume, J., N. Eckert, G. Chambon, M. Naaim, and L. Bel, 2013: Mapping extreme snowfalls in the french alps using max-stable processes. *Water Resour. Res.*, **49**, 1079–1098.

Gaume, J., J. Schweizer, A. van Herwijnen, G. Chambon, B. Reuter, N. Eckert, and M. Naaim, 2014: Evaluation of slope stability with respect to snowpack spatial variability. *J. Geophys. Res.*, **119**(9), 1783–1789.

Gaume, J., A. van Herwijnen, G. Chambon, K. Birkeland, and J. Schweizer, 2015: Modeling of crack propagation in weak snowpack layers using the discrete element method. *The Cryosphere*, **9**, 1915–1932.

Gauthier, D., 2007: *A practical field test for fracture propagation and arrest in weak snowpack layers in relation to slab avalanche release*. PhD thesis, PhD Thesis, 302pp. Department of Civil Engineering, University of Calgary, Alberta.

Gauthier, D. and B. Jamieson, 2008: Evaluation of a prototype field test for fracture and failure propagation propensity in weak snowpack layers. *Cold Reg. Sci. Technol.*, **51**(2), 87–97.

Geldsetzer, T. and J. Jamieson, 2001: Estimating dry snow density from grain form and hand hardness. In *Proceedings International Snow Science Workshop, Big Sky, Montana, USA, 1-6 October 2000*, 121-127.

Heierli, J., K. Birkeland, R. Simenhois, and P. Gumbsch, 2011: Anticrack model for skier triggering of slab avalanches. *Cold Regions Science and Technology*, **65**(3), 372–381.

Heierli, J., P. Gumbsch, and M. Zaiser, 2008: Anticrack nucleation as triggering mechanism for snow slab avalanches. *Science*, **321**, 240–243.

Jamieson, J. and C. Johnston, 2001: Evaluation of the shear frame test for weak snowpack layers. *Ann. Glaciol.*, **32**, 59–69.

McClung, D., 1979: Shear fracture precipitated by strain softening as a mechanism of dry slab avalanche release. *J. Geophys. Res.*, **84(B7)**, 3519–3526.

Podolskiy, E., M. Barbero, F. Barpi, G. Chambon, M. Borri-Brunetto, O. Pallara, B. Frigo, B. Chiaia, and M. Naaim, 2014: Healing of snow surface-to-surface contacts by isothermal sintering. *The Cryosphere*, **8**(5), 1651–1659.

Reiweger, I., J. Gaume, and J. Schweizer, 2015: A new mixed-mode failure criterion for weak snowpack layers. *Geophys. Res. Lett.*, **42**(5), 1427–1432.

Reuter, B., J. Schweizer, and A. van Herwijnen, 2015: A process-based approach to estimate point snow instability. *The Cryosphere*, **9**, 837–847.

Schweizer, J., K. Kronholm, J. Jamieson, and K. Birkeland, 2008: Review of spatial variability of snowpack properties and its importance for avalanche formation. *Cold Reg. Sci. Technol.*, **51(2-3)**, 253–272.

Szabo, D. and M. Schneebeli, 2007: Subsecond sintering of ice. *Applied Physics Letters*, **90**(15), 151916.

Timoshenko, S. and J. Goodier, 1970: *Theory of Elasticity*, volume 37. McGraw-Hill, 888.

van Herwijnen, A., J. Gaume, E. Bair, B. Reuter, K. Birkeland, and J. Schweizer, 2016: Field method for measuring the effective elastic modulus and fracture energy of snowpack layers. *J. Glaciol.*, **In Press**.

van Herwijnen, A. and J. Heierli, 2010: A field method for measuring slab stiffness and weak layer fracture energy. *Proceedings of the International Snow Science Workshop, Lake Tahoe, CA, USA, 2010* 232–237.

van Herwijnen, A., J. Schweizer, and J. Heierli, 2010: Measurement of the deformation field associated with fracture propagation in weak snowpack layers. *J. Geophys. Res.*, **115**(F3).

---

## Editor Decision (ED1)

MS 300-323M
4800 Oak Grove Drive
Jet Propulsion Laboratory
Pasadena, CA 91109-8099, U.S.A.
Tel (818) 970-8032
email: eric.larour@jpl.nasa.gov

November 21, 2016

Dear authors:

Thank you for submitting your revision to the manuscript entitled " **Snow fracture in relation to slab avalanche release: critical state for the onset of crack propagation** " for publication in *The Cryosphere*.

Your manuscript received two reviews during the interactive discussion, all pointing out to major revisions being necessary and further review required. The referees rated good or excellent in all categories (Originality, Scientific Quality, Significance and Presentation Quality), which according to TC policies is needed to be accepted for publication. I agreed at the time with both reviewers and pushed for further review. The major points raised by both reviewers that still needed to be addressed included:

Reviewer #1 being concerned mostly with the model description, and some specific points that will be of interest to all readers (such as line 152, crack tip position), and will add considerable value to the manuscript. Given that one of the most important results (line 211) to the community depends on the model implementation significantly, most points raised by reviewer 2 were to be addressed.

Reviewer #2 was concerned with referencing previous PSTs that showed results of decreasing critical length with increasing slope angle dependence. This was in my opinion one of the most critical issues in the manuscript. I found reviewer #2's argument that the authors did not present any field evidence to support the modeled slope angle dependence quite convincing, and this point had be addressed thoroughly.

After further review, I have now received the inputs from reviewer #2 (reviewer #1 did not participate in this round) and from new reviewer #3. Both reviewers rated good or excellent in all categories, except presentation quality that was rated Poor by reviewer #2. This needs to be corrected.

Reviewer #2 is satisfied with the comments and modifications to the manuscript. He raises some technical points that should still be addressed concerning statements that might be construed as "not true". These points need to be addressed, but do not warrant further

review. In addition, he commented on the general level of English writing, which should be addressed further.

Reviewer #3 was brought in specifically to address the concerns of reviewer #2 on the issue of decreasing critical length with increasing slope angle dependence. He appeared satisfied (even impressed) with the way this was argumented in the modified manuscript. He however raised specific issues that should also be addressed prior to publication. Specifically, I would agree with his analysis that the modeling should be presented in a more sensible way, without necessarily being confrontational.

In view of these arguments, I am accepting the manuscript for publication subject to minor revisions, which I will review myself. I expect the points raised above to be addressed, and the manuscript to be thoroughly vetted for english grammar.

Sincerely yours,

Dr. Eric Larour,
Editor *The Cryosphere*